# Wavelength engineerable porous organic polymer photosensitizers with protonation triggered ROS generation

Jinwoo Shin [1,2,8], Dong Won Kang[1,8], Jong Hyeon Lim[3,8], Jong Min An[4,8], Youngseo Kim[1], Ji Hyeon Kim [1], Myung Sun Ji[1], Sungnam Park [1] ✉, Dokyoung Kim [4,5,6,7] ✉, Jin Yong Lee [3] ✉, Jong Seung Kim [1] ✉ & Chang Seop Hong[1] ✉

Engineering excitation wavelength of photosensitizers (PSs) for enhanced reactive oxygen species (ROS) generation has inspired new windows for opportunities, enabling investigation of previously impracticable biomedical and photocatalytic applications. However, controlling the wavelength corresponding to operating conditions remains challenging while maintaining high ROS generation. To address this challenge, we implement a wavelength-engineerable imidazolium-based porous organic photocatalytic ROS generation system (KUP system) via a cost-effective one-pot reaction. Remarkably, the optimal wavelength for maximum performance can be tuned by modifying the linker, generating ROS despite the absence of metal ions and covalently attached heavy atoms. We demonstrate that protonated polymerization exclusively enables photosensitization and closely interacts with oxygen related to the efficiency of photosensitizing. Furthermore, superior tumor eradication and biocompatibility of the KUP system were confirmed through bioassays. Overall, the results document an unprecedented polymerization method capable of engineering wavelength, providing a potential basis for designing nanoscale photosensitizers in various ROS-utilizing applications.

Reactive oxygen species (ROS) are one of the most common chemicals in living organisms and industrial fields that respond to normal cellular functioning and catalytic process[1,2]. ROS can include chemically reactive radicals (type I), such as superoxide ($O_2^{\cdot-}$) or hydroxyl radicals, and molecular oxygen-driven non-radical molecules (type II), such as singlet oxygen ($^1O_2$)[3]. With a focus on their characteristic reactivity, numerous ROS-related studies have been reported in chemical and biological research for therapeutic and photocatalytic applications[4,5]. For example, photodynamic therapy (PDT) is one of the representative non-invasive remedies that attack malignant cells through ROS-induced cell death mechanisms[3]. In addition, ROS can be harnessed in various photocatalytic applications, including chemical bond generation[6–8], photoelectrolytic reduction[9,10], and photolysis, which can detoxify chemicals harmful

[1]Department of Chemistry, Korea University, Seoul 02841, Republic of Korea. [2]Department of Chemistry, Sarafan ChEM-H Institute, and Stanford Cancer Institute, Stanford University, Stanford, CA 94305, USA. [3]Department of Chemistry, Sungkyunkwan University, Suwon 16419, Republic of Korea. [4]Department of Biomedical Science, Graduate School, Kyung Hee University, Seoul 02447, Republic of Korea. [5]Department of Anatomy and Neurobiology, College of Medicine, Kyung Hee University, Seoul 02447, Republic of Korea. [6]KHU-KIST Department of Converging Science and Technology, Kyung Hee University, Seoul 02447, Republic of Korea. [7]UC San Diego Materials Research Science and Engineering Center, 9500 Gilman Drive, La Jolla, CA 92093, USA. [8]These authors contributed equally: Jinwoo Shin, Dong Won Kang, Jong Hyeon Lim, Jong Min An. ✉e-mail: spark8@korea.ac.kr; dkim@khu.ac.kr; jinylee@skku.edu; jongskim@korea.ac.kr; cshong@korea.ac.kr

to humans, such as chemical warfare agents (CWA) and factory pollutants[11–15].

In general, ROS can be generated by transferring energy from excited photons of the photosensitizer (PS) and can be utilized in chemical reactions. Efficiency of a PS directly governs the amount and types of ROS generated. Therefore, the type and role of PSs are quite important in pertinent reactions. In particular, PS with a peculiar wavelength according to specific activation conditions, such as deeply placed cancer or realistic conditions like sunlight, is needed but engineering the corresponding wavelength while maintaining high ROS generation efficiency is very challenging. This is because ROS generation relies heavily on the slight differences in energy levels induced by the structure of the PSs. Thus, it is difficult for conventional PSs to simply adjust their operating wavelengths, so their use is limited because new PSs that meet each condition must be newly synthesized and prepared (Fig. 1a). Therefore, a sophisticated and uncomplicated synthetic process is required[16,17].

Recently, porous materials such as metal-organic frameworks (MOFs), covalent-organic frameworks (COFs), and porous organic polymers (POPs) have been reported as novel classes of nano-photosensitizers[18–21]. Interestingly, their tailorable structure can absorb the light of various wavelengths, depending on the change of the organic linker to the framework[22], and their well-developed porous environment can accelerate the rapid transport of ROS whose lifetime is too short to activate where it is intended[23]. In particular, nanoscale POPs are expected to have the combined advantages of low-toxicity molecular organic photosensitizers and nano-sized metallic nano-particles with larger external surface areas, improved permeability, and increased retention effects. However, most of the reported POP-based photosensitizers are mainly composed of boron-dipyrromethene or porphyrin cores, which are well known for their $^1O_2$ generation motif but exhibit heavy atomic effects, such as metal ion loading and covalent bonding with halogens[24–26]. Despite their excellent $^1O_2$-generating ability, the toxicity of metal ions is a

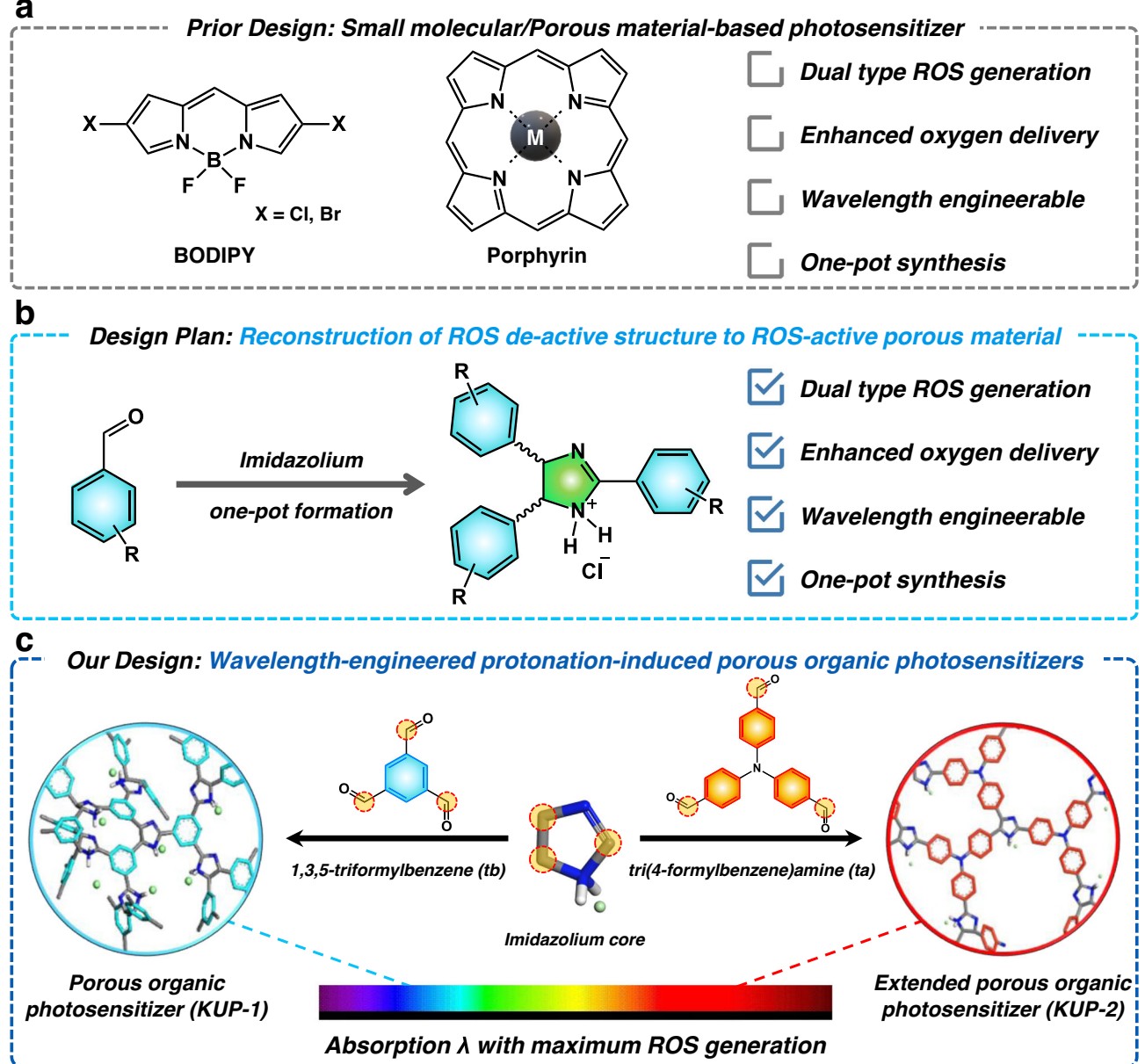

**Fig. 1 | Developing a novel class of photosensitizing agents. a** Previously reported conventional PSs. **b** Current design plan for promoting ROS-active porous material. **c** Schematic illustration of porous organic photosensitizers (**KUP-1**) and its extended version (**KUP-2**). The protonated imidazoline cores of porous polymers were formed from each aldehyde group of 1,3,5-triformylbenzene (tb) and tri(4-formylbenzene)amine (ta) through a cost-effective one-pot reaction.

formidable obstacle for practical applications in humans and ecosystems. To overcome these fundamental limitations, it is highly sought to discover a new class of PSs based on heavy atom-free POPs that are environmentally friendly and biocompatible.

Herein, we report a wavelength-engineered imidazolium-based porous organic polymeric photocatalytic ROS generation system (Korea University Porous Organic Polymer-1, **KUP-1** and its extended version, **KUP-2**), which has never been used in ROS-utilizing applications, including photocatalysis and therapy (Fig. 1b and c). Two POPs were synthesized via a cost-effective and scalable one-pot reaction without an additional catalyst. These POPs have excellent wettability and dispersibility in water because of the charged component of the structure, and they can load oxygen into their pores by virtue of their porosity and positive surface charge. When POPs are irradiated with light, the polymers can effectively generate ROS containing $^1O_2$ and $O_2^{\cdot-}$, which follow both ROS generation mechanism types I and II, despite the absence of covalently attached heavy atoms and light irradiation with a weak power of $1\,mW\,cm^{-2}$. Note that the wavelength for ROS generation is adjustable in the visible range. We applied state-of-the-art computational methods to this polymeric system, along with experimental evidence, to understand the mechanism of ROS generation, which elucidates the intersystem crossing (ISC) dominance by protonated polymerization. Finally, biological experiments demonstrated the biocompatibility of these materials. This system can be harnessed in various chemical and biological fields, such as photocatalysts and PDT agents.

## Results

### Synthesis and structural characterization

Porous organic polymeric photosensitizing agents (**KUP-1** and **KUP-2**) were prepared by the reaction of 1,3,5-triformlybenzene (tb) or tris(4-formylphenyl)amine (ta) with ammonium chloride in *N,N*-dimethylformamide (DMF) at 150 °C for 5 days via a one-pot reaction (Fig. 2a). This method allowed for the facile and scalable production of photosensitizers (Supplementary Fig. 1). The amorphous phase of the solids was confirmed by powder X-ray diffraction patterns, where no recognizable peaks were observed (Supplementary Fig. 2). Infrared (IR) spectra were used to identify the functional groups in the polymeric frameworks (Supplementary Fig. 3). After each one-pot reaction, new peaks at 3375 and $1652\,cm^{-1}$ appeared in the IR spectrum of **KUP-1**, while similar peaks were observed at 3416 and $1655\,cm^{-1}$ for **KUP-2**, which could be attributed to the N–H stretching of the ammonium group and –C=N– vibration, respectively. An additional N–H bending peak was observed at $1599\,cm^{-1}$ in the IR spectra of **KUP-2**. The data thus indicated the presence of protonated imidazoline (i.e., imidazolium) moieties in the framework. Detailed information on the atomic composition of the compounds was obtained using X-ray photoelectron spectroscopy (XPS), as shown in Fig. 2b and Supplementary Fig. 4. The nitrogen content in **KUP-2** was higher than that in **KUP-1** owing to the difference in the nitrogen content of each starting material's composition. Interestingly, the presence of chloride was commonly corroborated in the XPS survey scans of **KUP-1** and **KUP-2**. We suggested that the chloride attached to the protonated nitrogen in the imidazolium core act as a counter-anion. The XPS narrow scan of the N1*s* peak was conducted to investigate the chemical environment of nitrogen (Fig. 2c and Supplementary Fig. 5). The N1*s* peak of **KUP-1** (**KUP-2**) can be deconvoluted to three subpeaks centered at 401.02 (399.67), 399.30 (399.07), and 398.60 (398.37) eV, which correspond to the binding energies of $N^+$–H, C–N–C, and C–N=C, respectively[27,28]. This observation indicates two different chemical environments (*sp*$^2$ C–N and *sp*$^3$ C–N states) of nitrogens in the framework. In fact, two recognizable peaks in the solid-state $^{15}N$ NMR ($^{15}N$ ssNMR) data of **KUP-1** were observed at 43.58 and 123.78 ppm, revealing nitrogens of quaternary ammonium and imine, respectively (Fig. 2d). The corresponding peaks were also found at 35.75 and 123.08 ppm in the

spectrum of **KUP-2** (Supplementary Fig. 6). These results consistently suggest the formation of imidazolium within the framework. Additionally, for **KUP-2**, a peak at 102.81 ppm corresponded to nitrogen connected to three benzene rings. The detailed structure of the solids was analyzed using $^{13}C$ ssNMR spectroscopy, as shown in Supplementary Fig. 7 (the peak assigned number is shown in Supplementary Fig. 1). Distinct peaks (1) and (4–5) can be assigned to the carbon of the imine bond and the carbons in the imidazoline ring connected to benzene rings, respectively. Peak (2) can be attributed to the carbons of aromatic rings connected to the imidazoline ring, and a broad peak (3) in the range of 120–140 ppm can be assigned to the other carbons of the aromatic rings. These data support the proposed structure of each POP described in Fig. 1.

Thermogravimetric analysis (TGA) was conducted in the temperature range of 25–900 °C, under $N_2$, to inspect the thermal stability of POPs (Supplementary Fig. 8). The framework stabilities of both **KUP-1** and **KUP-2** were maintained up to 170 °C, implying that the materials are thermally stable in the operating temperature region of general remedial and catalytic applications. To check the porosity of POPs, we collected $N_2$ and $CO_2$ isotherms at 77 and 195 K, respectively, after degassing the samples at 120 °C for 10 h (Fig. 2e and Supplementary Fig. 9). The Brunauer–Emmett–Teller surface area was calculated as 19 and $29\,m^2\,g^{-1}$ from the $N_2$ isotherms at 77 K, and 173 and $184\,m^2\,g^{-1}$ from the $CO_2$ isotherms at 195 K for **KUP-1** and **KUP-2**, respectively. Thus, the data suggest that the KUP system has a wide range of pore-size distributions. Interestingly, **KUP-2** exhibited a higher surface area than **KUP-1** owing to the use of the extended organic monomer. The results can be associated with a well-packed framework by hydrogen bonds forming narrow pores, as confirmed by the broad peaks at ~$3300\,cm^{-1}$ in the IR spectra (Supplementary Fig. 3)[27]. The water droplet test exhibited no contact angle, revealing the hydrophilic nature of the frameworks (Supplementary Fig. 10). We suggest that the hydrophilic nature of the polymer originated from the charged quaternary ammonium and the high nitrogen content in the imidazolium core. To support our suggestion, the zeta potential of **KUP-1** was measured to be 15 mV, as shown in Fig. 2f. This feature can improve the degree of dispersion in water, which is highly advantageous for therapeutic and catalytic applications under aqueous conditions[29]. From scanning electron microscope (SEM) and tunneling electron microscope (TEM) images (Fig. 2g and Supplementary Figs. 11–14), the particle size distributions of spherical POPs were estimated to be 50–150 nm (**KUP-2** had relatively smaller particle sizes than **KUP-1**), which is suitable for ROS-utilized applications[27,30].

### Solution-based photophysical assays

To scan the absorbance region of the KUP systems, we performed solid-state UV–Vis (ssUV) spectroscopy (Supplementary Fig. 15). Strong absorbance bands were commonly observed in the visible range and tails up to 600 nm. Interestingly, the maximum absorbance peak range of **KUP-1** was at 200–400 nm, but the peak range shifted to longer wavelengths (250–500 nm) in the extended system. Thus, the full UV–Vis range of imidazoline-based POPs is available, and the maximum absorbance range can be designed simply by extension of the framework.

Based on the absorbance spectra, the $^1O_2$ generation ability of the KUP system was tested using 9,10-anthracenediyl-bis(methylene) dimalonic acid (ABDA), in phosphate-buffered saline (PBS) solution (Supplementary Figs. 16–19). After $0.2\,mg\,mL^{-1}$ of **KUP-1** was well-dispersed in the solution after sonication for 1 min, the resulting solution was irradiated at 430 nm using a xenon lamp with only weak power ($1\,mW\,cm^{-2}$) (Supplementary Fig. 16a). The initial intensity of the absorbance peak significantly decreased, reaching half the intensity only after 10 min. This phenomenon exhibits the exceptional ability of the KUP system to generate $^1O_2$ compared to conventional

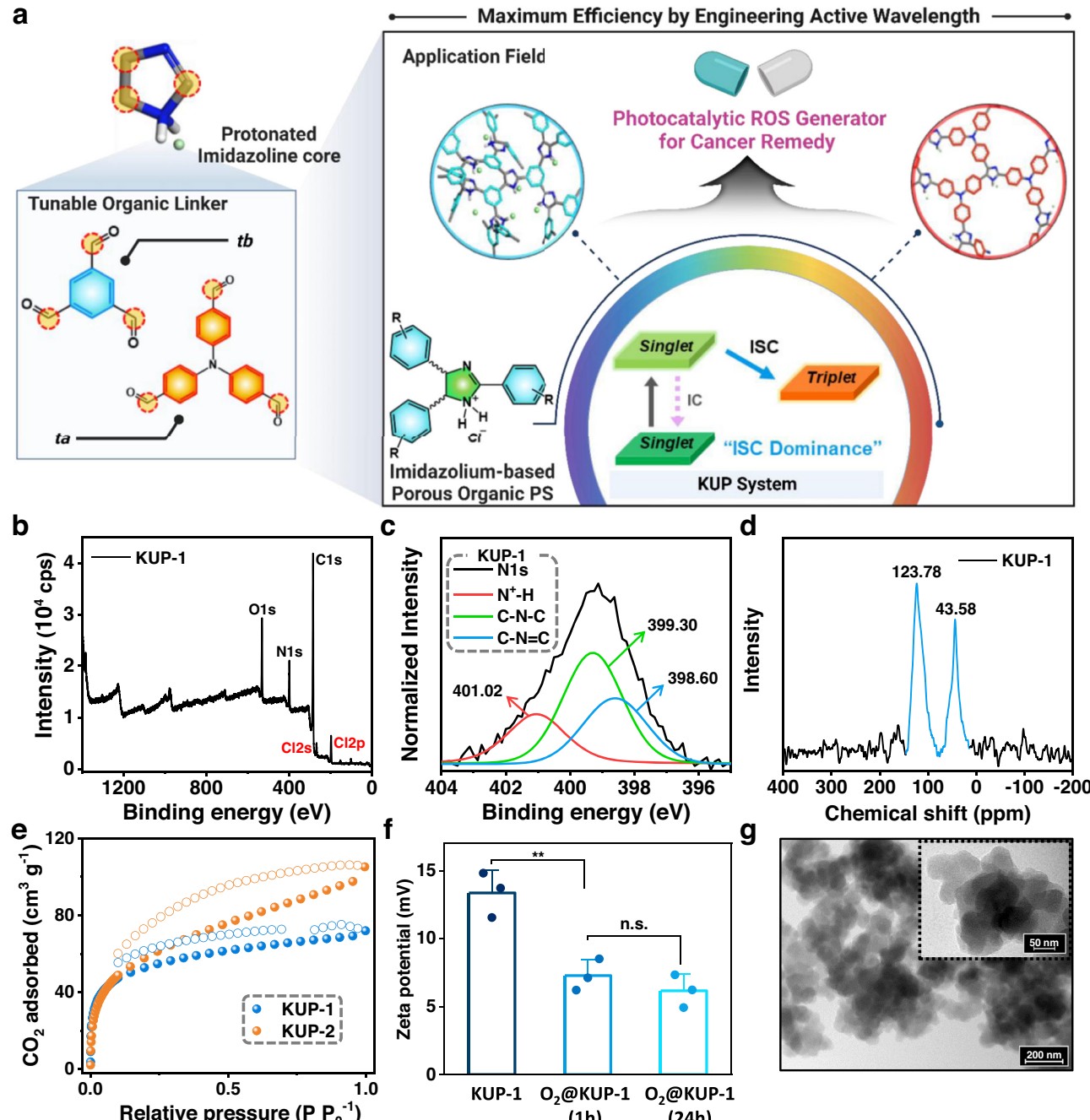

**Fig. 2 | Synthesis and structural characterization of porous organic photosensitizers. a** Schematic illustration of the porous organic polymeric photocatalytic ROS generation system, KUP system, and underlying imidazolium-based porous organic photosensitizers to engineering operating wavelength for maximum efficiency. **b** XPS survey scan and **c** narrow scan data of N1s peak of **KUP-1**. **d** Solid-state $^{15}N$ NMR data of **KUP-1**. **e** $CO_2$ isotherms of porous organic polymers at 195 K. **f** Surface charge distributions of **KUP-1** and **O$_2$@KUP-1**. The error bar represents mean ± SD ($n = 3$) with One-way ANOVA, Turkey's multiple comparisons tests; $F(2,6) = 24.33, P = 0.0013$; **KUP-1** vs. **O$_2$@KUP-1**) (1 h), $^{**}P = 0.0037$ (Mean Diff.: 6.080, 95.00% CI of diff.: 9.487), $^{**}P < 0.01$ and n.s. = non-significant. The dot plots represent Jitter points. **g** TEM images (×63,000 and ×250,000; inset) of the prepared **KUP-1**. The scale bars are 200 and 50 nm, respectively.

photosensitizers that normally produce $^1O_2$ under light irradiation, with powers >100 mW cm$^{-2}$, in therapeutic and photocatalytic applications[31–33]. We used additional excitation wavelengths of 660 and 808 nm to examine the generation capability of $^1O_2$ as a function of wavelength (Supplementary Fig. 16b and c). In both spectra, the $^1O_2$ generation performance of **KUP-1** was reduced (Supplementary Fig. 16d). To confirm the role of excitation light in the generation of $^1O_2$, we conducted the same absorbance experiments in the absence of a xenon lamp as a control group, and the decrease in intensity was not observed in the absorbance peak. $^1O_2$ generation did not occur in

either case of light irradiation without **KUP-1** and light irradiation with the starting material tb only (Supplementary Fig. 17b and c). These results indicate that both **KUP-1** and light irradiation are essential components to produce $^1O_2$, revealing that the generation ability originates from the polymeric framework of **KUP-1**.

Similarly, we performed the same photophysical experiments using **KUP-2** to demonstrate $^1O_2$ generation, resulting in noteworthy improvement in $^1O_2$ production at longer wavelengths such as 660 and 808 nm (Supplementary Fig. 18). Moreover, the initial intensity of the ABDA absorbance peak was reduced to half the intensity only

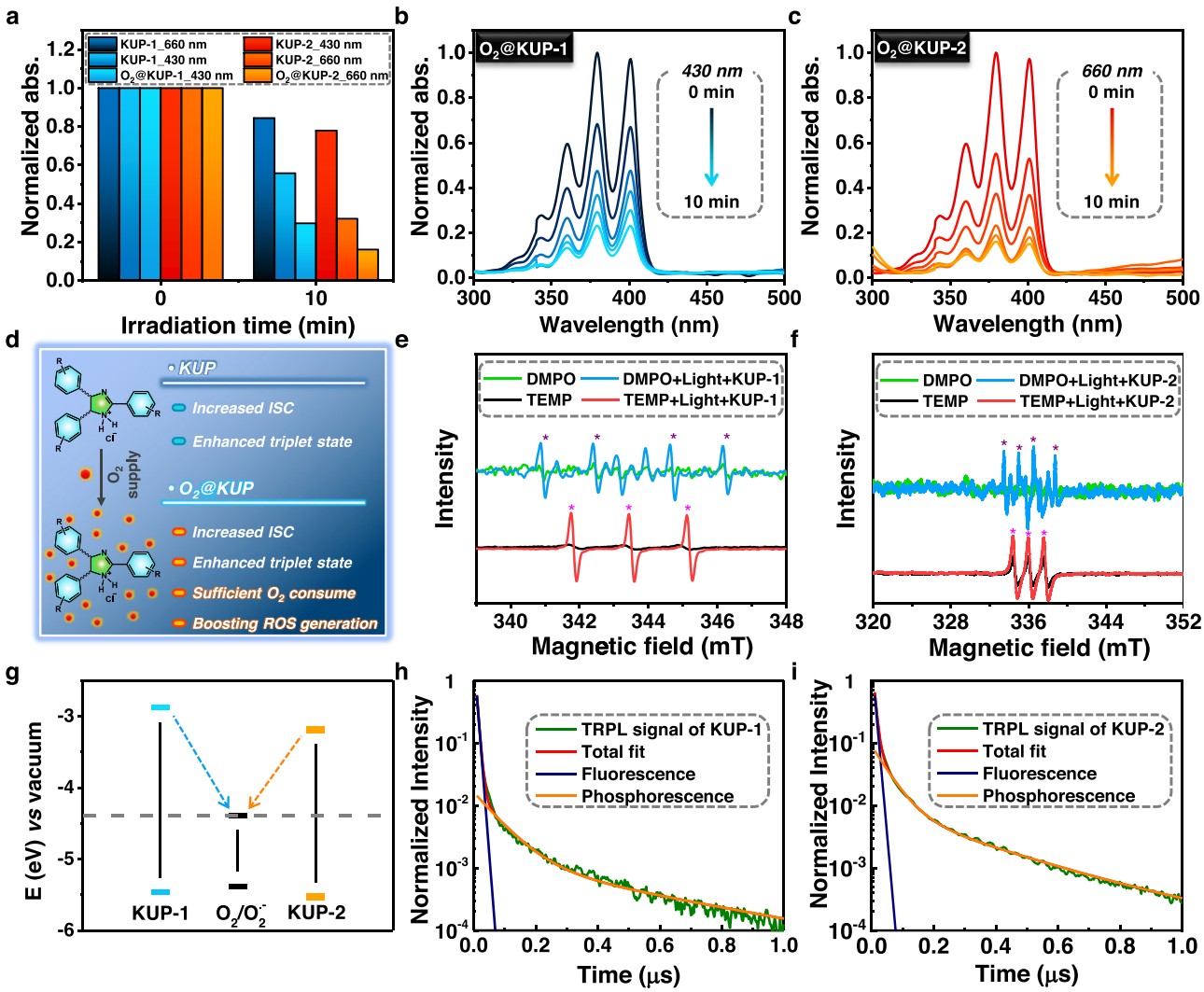

**Fig. 3 | Photophysical properties and ROS generation ability of porous organic photosensitizers. a** Normalized absorbance intensity comparison of experiments using ABDA indicator for $^1O_2$ detection from Fig. 2b, c, S16a, b, S18a, and b. **b** Time-dependent UV−Vis absorbance spectra of ABDA (100 μM) in PBS solution upon irradiation at 430 nm with a xenon lamp (1 mW cm$^{-2}$) in the presence of $O_2$-saturated **KUP-1** (**O₂@KUP-1**). **c** Time-dependent UV−Vis absorbance spectra of ABDA (100 μM) in PBS solution upon irradiation at 660 nm with a xenon lamp (1 mW cm$^{-2}$)

in the presence of $O_2$-saturated **KUP-2** (**O₂@KUP-2**). **d** Schematic illustration of the advantages of the KUP system with oxygen saturation. EPR spectra of **e KUP-1** and **f KUP-2** after irradiation with white light (purple and magenta asterisks indicate $O_2^{·-}$ and $^1O_2$, respectively). **g** Band gap diagram of KUP system. TRPL signal of **h KUP-1** and **i KUP-2** decomposed into fluorescence decay on the nanosecond timescale and phosphorescence decay on the microsecond timescale.

after 3 min, faster than that of **KUP-1**. The excellent $^1O_2$ generation arises from the extended polymeric framework of **KUP-2**, as verified by control group experiments (Supplementary Fig. 19). From these absorbance data, the wavelength suitable for $^1O_2$ generation can be easily tuned by the length of the linker in the construction of the framework, providing a new, important strategy for photosensitizer design. It is very attractive that this design strategy for wavelength tuning does not significantly affect physical properties such as particle size (Fig. 2g and Supplementary Figs. 11–14), which makes it suitable for ROS-harnessed application fields. We also examined the $^1O_2$ generation capabilities of $O_2$-loaded solids, **O₂@KUP-1** and **O₂@KUP-2**. The solids showed an overwhelming ability to generate $^1O_2$, as indicated by the reduced strength of ABDA (Fig. 3a−c). We expected that the protonated-KUP system with porosity and positive surface charge could effectively interact with negative oxygen molecules[34]. After $O_2$ loading, the surface charge of **KUP-1** decreased from 15 to 7.5 mV as determined by zeta potential measurements (Fig. 2f). To determine the oxygen capacity of KUP systems as carriers in aqueous, dissolved oxygen (DO) levels were

measured (Supplementary Fig. 20). Before measuring the DO level, $O_2$ gas from the balloon was transferred to the pores of the degassed KUP systems to obtain oxygen-impregnated solid **O₂@KUP-1** and **O₂@KUP-2**. Each solid was carefully soaked in water at a concentration of 0.1 mg mL$^{-1}$. As a result, we observed an increase in dissolved oxygen levels (-3.8 and -4.2 mg L$^{-1}$) higher than $O_2$-containing water without KUP systems (-1.1 mg L$^{-1}$). The difference between oxygen levels with/without KUP systems was calculated to be 2.7 mg L$^{-1}$ for **KUP-1** and ~3.1 mg L$^{-1}$ for **KUP-2**. This value is significantly higher than that of $O_2$ self-sufficient fluorinated polypeptide PHFB nanoparticles (-575 μM at 3 mg mL$^{-1}$)[35]. In addition, we measured the zeta potential after incubation of **O₂@KUP-1** in an aqueous solution to check the $O_2$ transport performance of the KUP system. As a result, the surface charge of **O₂@KUP-1** is still kept even after 24 h, indicating that oxygen content in **O₂@KUP-1** will not drop significantly during $O_2$ delivery (Fig. 2f). Thus, these data imply that the porous polymeric framework with a positive surface charge can serve as an oxygen carrier and promote an increase in the amount of oxygen dissolved in water, resulting in a large amount of

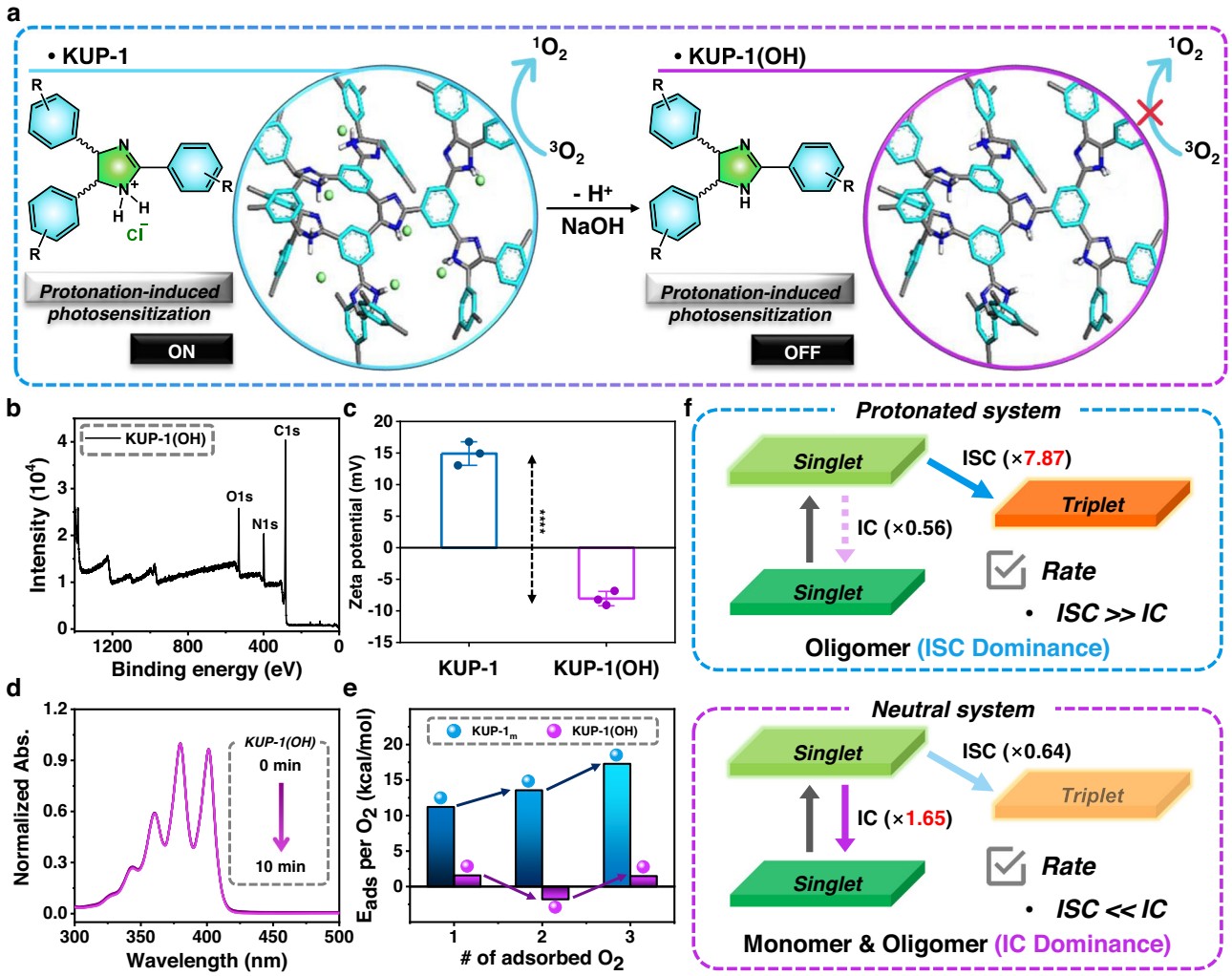

**Fig. 4 | ROS generation mechanism study using the imidazolium-based photosensitizing system and adsorption energy calculation between KUP system and O₂. a** Schematic illustration using optimized oligomeric structures of **KUP-1** and **KUP-1(OH)** with seven units of each monomer. The polymeric framework of **KUP-1** was optimized without counter anions and is called to **KUP-1ₘ** for computational convenience. **b** XPS survey scans data of **KUP-1(OH)**. **c** Surface charge distribution of **KUP-1(OH)**. Positive surface charge of **KUP-1** was changed to a negative charge after hydroxide ion treatment. The error bar represents mean ± SD

($n = 3$) with unpaired $t$-test; **KUP-1** vs. **KUP-1(OH)**, ****$P < 0.0001$ (Two-tailed, $t = 18.18$, df = 4). **d** Time-dependent UV–Vis absorbance spectra of ABDA (100 μM) in PBS solution upon irradiation at 430 nm with a xenon lamp (1 mW cm⁻²) in the presence of **KUP-1 (OH)**. **e** Adsorption energy calculation of **KUP-1ₘ** and **KUP-1(OH)** with the increasing number of oxygen molecules, respectively. **f** Schematic illustration of IC and ISC population changes in protonated and neutral systems before and after oligomerization based on DFT calculations.

ROS generation (Fig. 3d). A higher amount of ROS would enable elevated catalytic performance or treatment of cancer cells under hypoxic conditions[36].

To determine the different ROS types generated by the KUP system under light irradiation, electron paramagnetic resonance (EPR) measurements were performed with 2,2,6,6-tetramethylpiperidine (TEMP) and 5,5-dimethyl-1-pyrroline N-oxide (DMPO) as a ¹O₂ and O₂·⁻ generation indicator, respectively (Fig. 3e and f). When each KUP system in the TEMP solution was irradiated with white light, distinct peaks were observed, indicating the generation of ¹O₂. This result conforms to the UV–Vis absorbance experiments with ABDA indicator, suggesting a type II mechanism of ROS generation[5]. Interestingly, when each KUP system was irradiated with white light in DMPO solution, O₂·⁻ peaks were observed in the EPR spectra, corresponding to the type I mechanism of ROS generation[37]. Unlike ¹O₂, largely affected by the concentration of dissolved oxygen, other radicals, including O₂·⁻ exhibit excellent therapeutic effects under hypoxic conditions of cancer stem cells due to its O₂-less dependent generation mechanism[38]. The radicals have also been utilized to decompose various organic pollutants in aqueous environments[31].

To understand the ROS generation mechanism, the band gap of each KUP system in Fig. 3g was determined using a Tauc plot converted from the ssUV–Vis spectrum (Supplementary Fig. 21). The calculated band gaps were 2.58 eV for **KUP-1** and 2.34 eV for **KUP-2**. The potential level of the KUP systems was estimated using Ag/AgCl as the reference electrode (Supplementary Fig. 22). As a result, the representative valence band levels of POPs were evaluated as −5.46 eV (**KUP-1**) and −5.49 eV (**KUP-2**) vs. vacuum levels from the oxidation data. From these values, a schematic band diagram of the KUP systems is shown in Fig. 3g, and the ROS generation mechanism proposed is shown in Fig. 4. When the KUP systems are excited by light, light-induced electrons are generated in the conduction band and transferred to dissolved O₂, which is converted into O₂·⁻[39].

ROS generation through every photosensitization path is a phenomenon that is taken only when population transfer from singlet state to triplet state is accompanied. Therefore, we measured wavelength-dependent time-resolved photoluminescence (TRPL) signal to demonstrate the existence of triplet state population on KUP systems. As shown in Supplementary Fig. 23a and b, it was confirmed

that the TRPL signal of **KUP-2** was obtained at a longer wavelength overall than **KUP-1** when the TRPL signals were measured at 78 K. To measure the triplet state lifetime of the **KUP-1** and **KUP-2**, we compared the **KUP-1** TRPL signal at 530 nm and the **KUP-2** at 550 nm by plotting. Thus, we confirmed that the lifetime of the **KUP-2** is longer than that of the **KUP-1** (Supplementary Fig. 23c). The fluorescence and phosphorescence spectrum of each sample was extracted from the corresponding TRPL signal, and it was demonstrated that ISC was generated in **KUP-1** and **KUP-2**, respectively, by confirming the existence of the phosphorescence spectrum (Supplementary Fig. 24). Furthermore, fluorescence and phosphorescence signals of **KUP-1** and **KUP-2** were obtained by separating the TRPL signals, respectively (Fig. 3h and i). Thus, the data indicate direct experimental evidence of the formation of a triplet state (i.e., ISC) of the KUP system.

### Mechanistic study of ROS generation

To gain insight into the role of the charged section in the imidazoline core with respect to photosensitization of $O_2$, the **KUP-1** powder was soaked in a 1 M NaOH aqueous solution and stirred at 50 °C for 12 h (Fig. 4a). The resultant powder was washed with deionized water to provide the non-protonated porous organic photosensitizer, **KUP-1(OH)**. The IR spectrum of **KUP-1(OH)** showed that the broad band associated with N–H stretching around 3375 cm$^{-1}$ was significantly weakened compared with the IR peak of **KUP-1**. On the contrary, the other peaks in the fingerprint region for **KUP-1(OH)** were almost identical to those for **KUP-1** (Supplementary Fig. 25). Meanwhile, peaks related to chloride ions were not observed in the XPS survey scan of **KUP-1(OH)** (Fig. 4b). In addition, although the surface of **KUP-1(OH)** was measured as a negative charge (Fig. 4c), the solid-state NMR data confirmed that there was no significant change in the chemical environment after NaOH treatment (Supplementary Fig. 26). We drew a schematic band diagram in Supplementary Fig. 27 from ssUV, Tauc plots, and cyclic voltammetry (CV) data, showing a band gap of 2.75 eV for **KUP-1(OH)**, with a representative valence band level of −5.54 eV vs. vacuum level. This value is similar to that of **KUP-1**. However, the $^1O_2$ generation test of **KUP-1(OH)** under the same experimental procedure led to an absolute suppression of the performance, as shown in Fig. 4d. Thus, we postulated that the charged component of **KUP-1** played an essential role in the generation of $^1O_2$.

To understand the difference in the $^1O_2$ generation ability of **KUP-1** and **KUP-1(OH)** via electron transfer, the excited-state dynamics should be investigated. Therefore, we performed density functional theory (DFT) calculations to support the reason behind the comparison experiment results on $^1O_2$ generation using **KUP-1** and **KUP-1(OH)**. After the photosensitizer is excited to the $S_1$ state by the irradiated light, the excited system can be relaxed through a non-radiative process such as internal conversion (IC) and ISC processes. These two processes are competitive, and their relative relationship affects the electron population in the $T_1$ state, which is an important factor for $^1O_2$ generation due to the intrinsic triplet multiplicity of the oxygen molecule. As a result, we calculated and compared the IC and ISC rates of the excited **KUP-1** and **KUP-1(OH)** systems.

Since the ROS generation-capable KUP system is in polymeric frameworks, we demonstrated the changing trend of the relative excited-state dynamics with respect to the size of the system used in this study. We performed calculations for the **KUP-1** monomer and oligomer systems with seven monomer units due to the practical limitation of computational costs for polymer systems. For the **KUP-1(OH)** system, monomer and analogous oligomer systems were investigated. In the case of the **KUP-1** oligomer, positively charged model systems, excluding counter anions (model **KUP-1**, **KUP-1$_m$**), were calculated due to the impractical high computational cost for the oligomer **KUP-1** system, including counter anions. All the systems investigated are shown in Fig. 4a and Supplementary Fig. 28.

**Table 1 | Computational IC and ISC rate constant for KUP-1(OH), model KUP-1 (KUP−1$_m$), excluding counter anion, monomer and oligomer with 7 monomer units, and KUP-1 monomer including counter anion**

| | IC rate constant ($10^{12}$ s$^{-1}$) | | ISC rate constant ($10^{12}$ s$^{-1}$) | |
|---|---|---|---|---|
| | **Monomer** | **Oligomer** | **Monomer** | **Oligomer** |
| **KUP-1(OH)** | $1.70 \times 10^{-1}$ | $2.80 \times 10^{-1}$ | $4.19 \times 10^{-4}$ | $2.68 \times 10^{-4}$ |
| **KUP-1$_m$** | 1.51 | $8.50 \times 10^{-1}$ | $3.46 \times 10^{-7}$ | $2.72 \times 10^{-6}$ |
| **KUP-1** | $6.64 \times 10^{-1}$ | – | $6.64 \times 10^{-2}$ | – |

The ISC rate between $S_1$ and $T_1$ was computed from the semi-empirical Marcus theory (Eq. (1))[40].

$$k_{ISC} = \frac{4\pi^2}{h} \frac{\langle S_1|H_{SOC}|T_1 \rangle^2}{\sqrt{4\pi\lambda k_B T}} \exp\left(-\frac{(MECP)^2}{k_B T}\right) \quad (1a)$$

$$MECP = \frac{(\Delta E_{ST} - \lambda)^2}{4\lambda} \quad (1b)$$

where $h$, $\lambda$, $k_B$, $T$, $\langle S_1|H_{SOC}|T_1\rangle$, MECP are the plank constant, reorganization energy, Boltzmann constant, temperature, spin–orbit coupling (SOC) constant, and minimum energy crossing point respectively, which are listed in Supplementary Table 1. Here, MECP could be derived from the energy difference between the $S_1$ and $T_1$ states ($\Delta E_{ST}$) and reorganization energy, as shown in Eq. (1b). The calculated results for the IC and ISC rates of the optimized KUP systems are listed in Table 1. Although ISC rates are quantitatively slower than IC rates, based on the computational results, the differences between monomers and oligomers show that the changing direction of the IC and ISC rates originated from oligomerization are opposite in **KUP-1$_m$** and **KUP-1(OH)**. According to the calculated results, during oligomerization in **KUP-1(OH)**, the IC rate became faster (1.65 times) and the ISC rate became slower (0.64 times). However, during oligomerization in **KUP-1$_m$**, the IC rate became slower (0.56 times) and the ISC rate became faster (7.87 times). These tendencies for **KUP-1(OH)** and **KUP-1$_m$** might be larger in practical polymeric systems. Based on this expectation, contrasting dynamic changes indicate that the **KUP-1(OH)** polymer would have faster IC than the ISC process and that the **KUP-1** polymer would have comparable rates of IC and ISC processes. This suggests that the electronic population in the triplet state might accumulate only in **KUP-1**, resulting in exclusive $^1O_2$ generation on protonated KUP system. This agrees with the experimental results for the formation of the triplet state of **KUP-1** (Fig. 3h) and $^1O_2$ generation only by the protonated KUP systems (Figs. 3a and 4d). Furthermore, when we considered the counter-anion effect, **KUP-1** had a much faster ISC and slower IC rates than **KUP-1$_m$**, as shown in Table 1. This counter-anion effect also supports the distinguishable $^1O_2$ generation ability of **KUP-1** (Fig. 4f).

Furthermore, we confirmed how well the **KUP-1** system interacts with oxygen molecules by calculating the adsorption energy between them (Fig. 4e). Adsorption energy was calculated per oxygen molecule by using the following equation:

$$E_{ads} = -\frac{1}{n}[E_{system} - (E_{adsorbent} + E_{adsorbate})] \quad (2)$$

where $n$ is the number of oxygen molecules, and $E_{system}$, $E_{adsorbent}$, and $E_{adsorbate}$ are energies of oxygen adsorbed **KUP-1**, pristine **KUP-1**, and oxygen molecules, respectively. We optimized the oligomer system of **KUP-1$_m$** and **KUP-1(OH)** with oxygen molecules to obtain the adsorption energy per oxygen molecule according to the increasing number of that and compared variation of adsorption energies. As shown in

Fig. 4e and Supplementary Table 2, The adsorption energies per oxygen molecule on **KUP-1m** positively increase as the number of oxygen molecules increases (11.21, 13.56, and 17.28 kcal/mol for 1, 2, and 3 $O_2$ molecules, respectively) while the adsorption energies of **KUP-1(OH)** show relatively similar and smaller value (1.57, −1.79, and 1.48 kcal/mol for 1, 2, and 3 $O_2$ molecules), which indicates that the protonated KUP system might be more favorably adsorb $O_2$ molecules than neutral KUP(OH) system.

## In vitro photo-induced cytotoxicity of KUP system

Given that the KUP system has a high capability for oxygen due to the favorable adsorption toward $O_2$ molecules and outstanding ability to generate ROS, we investigated the cellular effects of **KUP-1** and **O₂@KUP-1**. First, the photo-induced cytotoxicity assay for U87MG was evaluated under irradiation at 530 nm after treatment with **KUP-1** and **O₂@KUP-1** for 12 h incubation. Next, the toxicity was evaluated at 30 h from the time of treatment. As expected from the preceding data, the cytotoxicity of U87MG increased with photoirradiation at 530 nm after treatment with **KUP-1** or **O₂@KUP-1**, compared with non-irradiated cells after treatment. Therefore, the cytotoxicity results indicate that **KUP-1** and **O₂@KUP-1** could be utilized as photosensitizers that exhibit off-on cytotoxicity in the absence and presence of photoirradiation at 530 nm (Fig. 5a). A noteworthy phenomenon was that the toxicity effect on U87MG tends to be enhanced in the case of the **O₂@KUP-1** at a high concentration (0.1–0.4 mg mL$^{-1}$) more than that of the **KUP-1** due to the increased generation of ROS via the number of oxygen molecules interacting with PS enhanced by high $O_2$ affinity. Hence, the results mean that the $O_2$ impregnated on **KUP-1** could boost $^1O_2$ generation and act as one of the $^1O_2$ generator ingredients, attracting the $^1O_2$-induced apoptosis pathway[41]. Unlike **KUP-1** and **O₂@KUP-1**, we could not observe the positive results using **KUP-2** and **O₂@KUP-2** in diverse cell lines (C6, HeLa, and U87MG) as shown in Supplementary Fig. 29. To visualize the cell death caused by **KUP-1** and **O₂@KUP-1** in the presence of photoirradiation at 530 nm, we used luciferase-containing U87MG cells (Luc-U87MG). Live-cell imaging analysis was performed after treatment with **KUP-1** or **O₂@KUP-1** at 12 h and photo-irradiated at 530 nm for 3 min. As seen in Fig. 5b, the luminescence from Luc-U87MG cells decreased in the group treated with **O₂@KUP-1**, compared with the group treated with **KUP-1**. In particular, the cytotoxicity of **O₂@KUP-1** was confirmed by live-cell imaging. After pretreatment with **O₂@KUP-1** for 12 h and post-treatment with 530 nm irradiation, the live-cell images showed a marked difference in terms of the number of live cells. Generally, lactate dehydrogenase (LDH), an intracellular enzyme, is released upon cell death (apoptosis or necrosis) due to the destruction of the cell membrane by ROS-induced phototoxicity[42]. As seen in Fig. 5c, the group treated with **KUP-1** or **O₂@KUP-1** did not differ from the untreated group, but interestingly, laser irradiation at 530 nm in these groups induced an increase in LDH release. These results thus support selective phototoxicity that **KUP-1** and **O₂@KUP-1** produce enough ROS to be toxic in a short time (exposure time <3 min). Based on these results, we next evaluated the hemolysis induced by **KUP-1** and **O₂@KUP-1** under irradiation at 530 nm (Fig. 5d). **KUP-2** and **O₂@KUP-2** were also evaluated by hemolysis tests under irradiation at 660 nm (exposure time: 5 min) (Supplementary Fig. 30). Hemolysis is an indicator of exposure to hemoglobin by the destruction of red blood cells (RBCs). Hemolysis tests for the KUP system were used to indicate the toxicity to cell membranes and potential toxicity to intravenous (i.v.) injection of the photosensitizers[43]. The results indicate that pretreatment with **KUP-1**, **O₂@KUP-1**, **KUP-2**, and **O₂@KUP-2**, along with post-irradiation (530 and 660 nm), showed negative hemolysis of RBCs, indicating high biocompatibility while having potential as photosensitizers. Based on these results, we conclude that the superior ROS-induced cancer cell eradication and biocompatibility of imidazolium-based porous organic photosensitizers

potentially provide new insights into the advantages of the photosensitizing system in future clinical fields.

## In vivo photo-induced tumor eradication of KUP system

Based on our strategy, readily found to suppress the tumor growth in vitro, we next verified the antitumor potency of the KUP system. The BALB/c mice were first gifted with a U87MG cell line on the dorsal side and their vital were monitored by measuring their body weight every 3–4 days. After 3 days, the tumor-inoculated mice were treated with **KUP-1** and **O₂@KUP-1**, **KUP-2**, and **O₂@KUP-2** with a cycle of 2 days of photo-induced treatment to confirm the possibility of inhibiting tumor proliferation (Figs. 5e, 4f, and Supplementary Fig. 31). Given that the photophysical ability of **KUP-1** could emit at red wavelengths (Supplementary Figs. 23 and 24), fluorescence tissue imaging system (FTIS) images confirmed the residual status of **KUP-1** and **O₂@KUP-1** on the same day when the light irradiation. As seen in Fig. 5g, **KUP-1** and **O₂@KUP-1** still remained in the tumor site and can be used as a photocatalytic ROS generator to induce cancer cell attenuation. Based on this result, we performed the irradiation process in 1 day after treating **KUP-1** or **O₂@KUP-1** (2 cycles). As a result, the tumor size was markedly diminished in the group of **KUP-1** and **O₂@KUP-1** with light irradiation, and no substantial body weight loss was observed during the whole therapeutic course (Fig. 5h). In addition, these results were similarly observed in experiments using **KUP-2** and **O₂@KUP-2** (Supplementary Fig. 32). The difference in our results between in vitro and in vivo might be caused by the limitations of 2D cultures, such as cell shape with forced polarity and lack of immune cells. The plate of 2D culture was modified to force polarity to make the cell easier to attach to the plate. The modified plate drives to change the cellular morphology, unlike in vivo, although it helps the cell to attach to the plate. Also, the tumor microenvironment consists of diverse cell types, even containing the normal cell and immune cells, which all have a role in the function of the tumor[44]. Thus, many drugs have worked differently in vitro and in vivo. Based on these things, **KUP-2** and **O₂@KUP-2** might work near the tumor site connecting with diverse cells in a 3D structure. Since selective tumor attenuation remains a significant problem in modern oncology, such a novel class of POP-based photosensitizers, the KUP system, potentially provides an appealing complementary strategy toward the non-invasive and effective photo-triggered therapeutic efficiency. Although the evaluation was performed at short-cycle of PDT (treatment and rest) and high concentration (8 mpk), **KUP-1** and **O₂@KUP-1** showed no remarkable toxicity symptoms in mice during the experimental period (16 days) (Fig. 5i). These results thus indicate that KUP system can confine the active photo-induced tumor ablation and attenuate off-target damage. To explore the tumor suppression effect underlying photo-induced tumor eradication of the KUP system, the protein profiling assay of **KUP-1** and **O₂@KUP-1** was performed by independent experiments with duplicate using the blood from GBM-xenograft model in 16 days (Fig. 5i and j). The result showed that the levels of C5/C5a and IL-6 were increased by the treatment of **KUP-1** and **O₂@KUP-1**, dependent on light irradiation in both cases. Additionally, protein levels of I-TAC and CD54 increased slightly in treatment with **O₂@KUP-1** under light irradiation, while IL-27 decreased. On the basis that I-TAC, C5/C5a, IL-6, CD54, and IL-27 are the cytokines related to the immune system, we conclude that the KUP system acts as ROS photogenerators and induces the immune activity to control the release of cytokines, which cause the tumor eradication. Overall, these promising results collectively demonstrated the achievement of our novel strategy using a new class of POP-based photocatalytic ROS generators (KUP system) for accurate tumor eradication and manifested that the newly introduced imidazolium-based photosensitizers (**KUP-1** and **KUP-2**) are state-of-the-art oxygen-appended PDT agents.

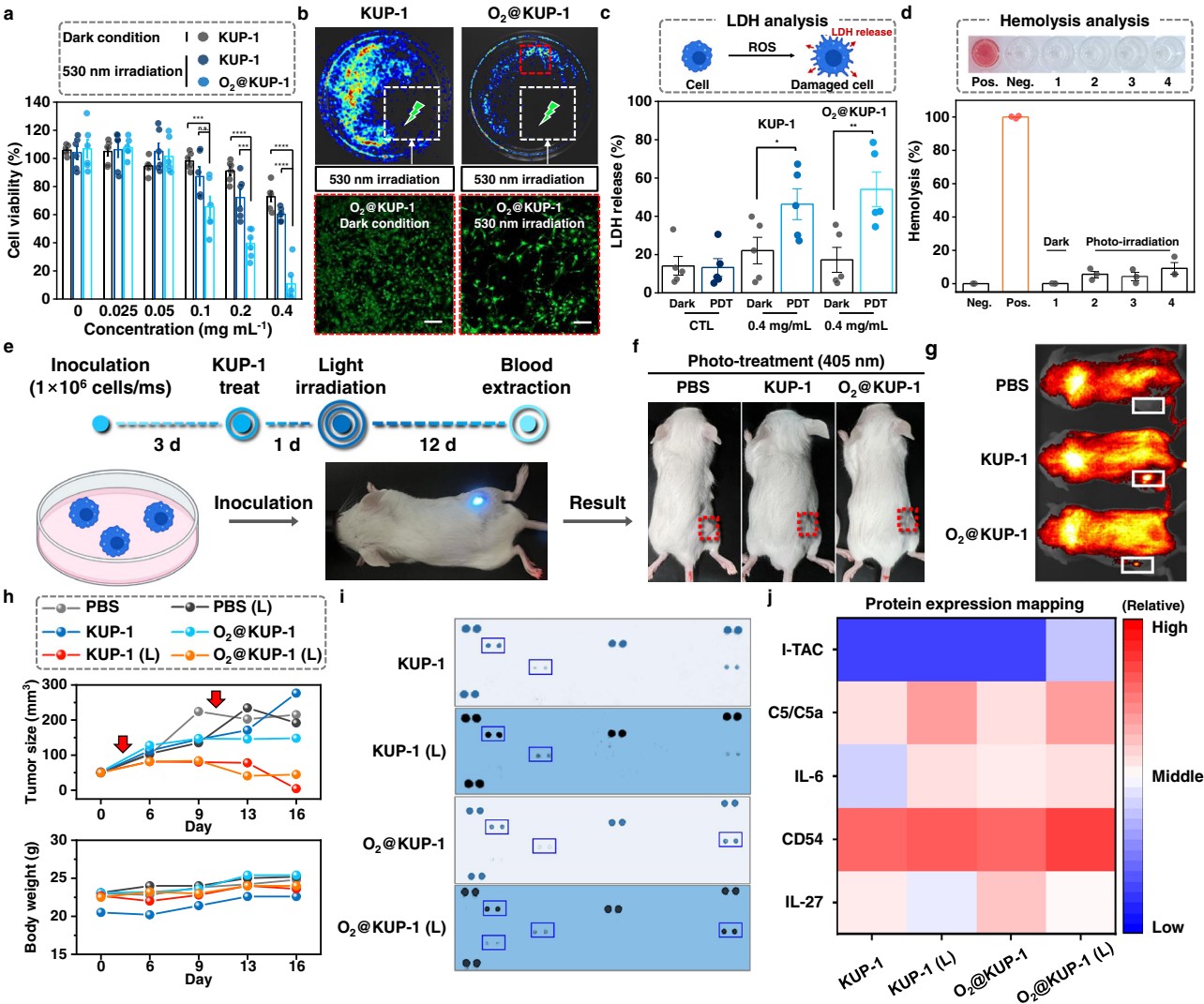

**Fig. 5 | In vitro and in vivo bioactivity of KUP-1 and O₂@KUP-1. a** Cytotoxicity assays of **KUP-1** and **O₂@KUP-1** with/without the presence of 530 nm laser irradiation. The error bar represents mean ± SEM ($n = 6$) with Two-way ANOVA, Turkey's multiple comparisons tests; Interaction: $F_{(10,90)} = 11.12$, $P < 0.0001$; Row factor: $F_{(2,90)} = 38.10$, $P < 0.0001$; Column factor: $F_{(5,90)} = 76.02$, $P < 0.0001$). 0.1 mg/mL **KUP-1** (Dark) vs. 0.1 mg/mL **O₂@KUP-1** (530 nm); ***$P = 0.0006$, 0.1 mg/mL **KUP-1** (530 nm) vs. 0.1 mg/mL **O₂@KUP-1** (530 nm); n.s., 0.2 mg/mL **KUP-1** (Dark) vs. 0.2 mg/mL **O₂@KUP-1** (530 nm); ****$P < 0.0001$, 0.2 mg/mL **KUP-1** (530 nm) vs. 0.2 mg/mL **O₂@KUP-1** (530 nm); ***$P = 0.0005$, 0.4 mg/mL **KUP-1** (Dark) vs. 0.4 mg/mL **O₂@KUP-1** (530 nm); ****$P < 0.0001$, 0.4 mg/mL **KUP-1** (530 nm) vs. 0.4 mg/mL **O₂@KUP-1** (530 nm); ****$P < 0.0001$. *$P < 0.05$, **$P < 0.01$, ***$P < 0.001$, ****$P < 0.0001$, and n.s. = non-significant. The dot plots represent Jitter points. **b** Live-cell images after treatment with **KUP-1** and **O₂@KUP-1** under photoirradiation at 530 nm. Scale bars: 200 μm. **c** Lactate dehydrogenase (LDH) assays after treatment of **KUP-1** and **O₂@KUP-1**. The error bar represents mean ± S.E.M. ($n = 5$) with Two-way ANOVA, Turkey's multiple comparisons tests; Interaction: $F_{(2,24)} = 3.906$, $P = 0.0340$; Row factor: $F_{(1,24)} = 12.89$, $P = 0.0015$; Column factor: $F_{(2,24)} = 6.454$, $P = 0.0057$). 0.4 mg/mL **KUP-1** (Dark) vs. 0.4 mg/mL **KUP-1** (530 nm); *$P = 0.0245$, 0.4 mg/mL **O₂@KUP-1** (Dark) vs. 0.4 mg/mL **O₂@KUP-1** (530 nm); **$P = 0.0037$. *$P < 0.05$, **$P < 0.01$, ***$P < 0.001$, ****$P < 0.0001$, and n.s. = non-

significant. The dot plots represent Jitter points. **d** Hemolysis tests for **KUP-1** and **O₂@KUP-1**. Neg: negative control (PBS), Pos: positive control; 0.1% (v/v) Triton X-100. 1: the group which KUP-1 treats without irradiation, 2: the group which is treated by PBS with irradiation at 530 nm, 3: the group which is treated by **KUP-1** with irradiation at 530 nm, 4: the group which is treated by **O₂@KUP-1** with irradiation at 530 nm. An inset photograph is a supernatant of the damaged red blood cells. The error bar represents mean ± SEM ($n = 3$) and the dot plots represent Jitter points. **e** Schematic illustration for in vivo evaluation of **KUP-1** with PDT irradiation in GBM-xenograft model. **f** Images of mouse condition after treatment of 1 × PBS, **KUP-1** (8 mpk; mg kg⁻¹), and **O₂@KUP-1** (8 mpk) with photoirradiation for 13 days. **g** FTIS images of the mouse with treatment of 1 × PBS, **KUP-1** (8 mpk), and **O₂@KUP-1** (8 mpk). **h** Tumor sizes and body weights of the mice in each tested group were recorded during treatment with/without 405 nm irradiation (75 mW cm⁻², 3 min) at the end-point. Red arrows indicate the PS treatment date. **i** Profiler mouse cytokine assay using extracted blood from the GBM-xenograft model at the end-point. Blue box means cytokine proteins that are increased after irradiation at 405 nm. **j** Protein expression mapping of **KUP-1** and **O₂@KUP-1**. The data was analyzed using (**i**). The number of mice per group is 6; independent experiments with duplicates repeatedly obtained all data. L: laser irradiation.

## Discussion

We prepared a wavelength-engineerable imidazolium-based porous organic polymeric photocatalytic ROS generator (**KUP-1**) and an extended version (**KUP-2**) through a cost-effective one-pot reaction, a new type of POP-based PS that has never been used in therapeutic and photocatalytic applications using ROS. A key design component

incarnated in the KUP system is the use of an appropriately modified organic linker to control the wavelength corresponding to specific operating conditions while maintaining high ROS generation efficiency and biocompatibility. Therefore, the KUP system can be harnessed with tunability in the visible range by simply modulating the components of PSs. Photophysical analysis of the KUP system proved

effective at promoting both type I and II ROS generation mechanisms under light irradiation, and remarkably, $^1O_2$ (type II ROS) was strongly generated under photoirradiation with weak power ($1\,mW\,cm^{-2}$). These results indicate that the KUP system enables the design of PS with peculiar excitation wavelengths according to specific activation conditions for optimal performance and type I and II photosensitizing, which are crucial factors in constructing PSs for various ROS-utilizing applications. We found that the charged component of the materials exhibited excellent wettability, dispersibility, and $O_2$ affinity that enhanced the amount of dissolved oxygen. Theoretical calculations suggest that the mechanism of ROS generation, only in the protonated system, is associated with ISC dominance triggered by the polymerization of a porous framework with charged moieties. In addition, the adsorption energy calculations on **KUP-1$_m$** and **KUP-1(OH)** with an increasing number of oxygen molecules presented that the protonated system is the key factor for high affinity with oxygen, which can influence the efficiency of $^1O_2$ generation. With such a unique photocatalytic ROS generation mode of action, excellent selective antitumor efficiency and non-invasive biocompatibility were elucidated through biological experiments. Furthermore, the protein profiling assay demonstrated the tumor suppression effect underlying photo-induced tumor eradication of the KUP system, which induces the immune activity to control the release of cytokines. Overall, these protonated POP-based photosensitizing systems, with tunable excitation wavelengths, could provide a potential basis for designing nanoscale porous organic photosensitizers for a variety of ROS-enabled applications.

## Methods

### Preparation
All starting chemicals and solvents for the synthesis were obtained from commercial suppliers (Merck, Samchun, TCI, Thermofisher) and used without further purification.

### Synthesis of KUP-1 and KUP-2
A 23 mL Teflon-lined cup was charged with tb (1.207 g, 7.45 mmol), ammonium chloride (2.39 g, 44.73 mmol), and DMF (10 mL). The cup was mounted in an autoclave and heated at 150 °C for 5 days. The precipitated solid was filtered and washed thoroughly with DMF, water, acetone, and methanol. The resultant pale-yellow **KUP-1** powder was dried at 100 °C under vacuum for 10 h. Yellowish **KUP-2** powder was prepared according to the same synthetic procedure as **KUP-1**, except that ta was used instead of tp. The elemental analysis of the sample (%) is as follows: Found for **KUP-1** (C, 55.35; H, 6.05; N, 12.98) and **KUP-2** (C, 67.20; H, 6.68; N, 13.20).

### Synthesis of KUP-1(OH)
A 70 mL vial was charged with ~100 mg of **KUP-1** and 50 mL of 1 M NaOH solution. After the vial was sealed, the mixture was stirred at 50 °C for 12 h. The solid was filtered and washed thoroughly with water, acetone, and methanol. The washed powder was dried at 100 °C in a drying oven for 12 h to obtain **KUP-1(OH)**.

### Solution test with UV−Vis absorbance measurement for $^1O_2$ detection
UV−Vis absorbance spectra were obtained using a Jasco V-750. The $^1O_2$ generation ability of POPs was assessed by UV−Vis absorbance spectra of ABDA (100 µM), a $^1O_2$ capture agent, in PBS solution (10 mM, containing 1% DMSO). For UV−Vis spectroscopy, POPs ($0.2\,mg\,mL^{-1}$) were sonicated for 1 min and dispersed well in PBS solution. The resulting solution was irradiated with a xenon lamp ($1\,mW\,cm^{-1}$) at a target wavelength for 10 min. The corresponding absorption spectra were recorded immediately after light irradiation. As a result of $^1O_2$ generation derived by POPs, the absorbance of ABDA was greatly reduced due to the oxidative decomposition effect of $^1O_2$ on ABDA.

## Computational calculations
Non-adiabatic molecular dynamics (NAMD) simulation based on the Kohn−Sham density functional theory (DFT) framework[45] was performed to calculate the IC rate using the PYXAID software package[46,47], which manages the motion of lighter electrons quantum mechanically and heavier nuclei classical mechanically. This package implements decoherence-induced surface hopping (DISH)[48] to describe the dynamics of charge in the excited state. The electronic state and adiabatic MD trajectories were obtained from ab initio molecular dynamics (AIMD), which were performed using the Vienna Ab initio Simulation Package (VASP)[49] based on the projector-augmented wave (PAW) pseudopotential theory, using a plane-wave basis set with Perdew, Burke, and Ernzerhof (PBE)[50]. The AIMD simulation was performed for 1 ps with a 1 fs time interval. The SOC constant between the $S_1$ and $T_1$ states was calculated using the Q-Chem 5.3 software packages[51], using the B3LYP hybrid functional[52] with the 6-31G(d) basis set. The system total energy, reorganization energy, $\Delta E_{ST}$, excitation energy, and oscillator strength were calculated using the Gaussian16 software package[53], using the B3LYP hybrid functional with 6-31G(d) basis sets. In this study, optimization and time-dependent DFT (TDDFT)[54,55] calculations were performed.

## Evaluation of the photo-induced cytotoxicity of KUP-1
U87MG cells ($5 \times 10^3$) were plated on a 96-well plate with a flat-bottom plate (SPL, Korea) for 24 h. When the confluency of cells reached 80%, the culture medium (Dulbecco Modified Eagle Medium containing 10% fetal bovine albumin and 1% penicillin−streptomycin) was changed to serum-free (SF) media to enhance the efficiency of the uptake of materials. After incubation for 30 min, the cells were treated with **KUP-1** and **O$_2$@KUP-1**. Irradiation at 530 nm ($60\,mW\,cm^{-2}$, 3 min) was performed after treatment with **KUP-1** and **O$_2$@KUP-1** for 12 h on fresh SF media. The phototoxicity of **KUP-1** and **O$_2$@KUP-1** was evaluated using a cell counting kit-8 (Dojindo, Japan), according to the manufacturer's protocols after incubation for 30 h.

## Live-cell images with KUP-1 and O$_2$@ KUP-1
To visualize the frequency of live cells after treatment with **KUP-1** ($0.4\,mg\,mL^{-1}$) and **O$_2$@ KUP-1** ($0.4\,mg\,mL^{-1}$) under irradiation at 530 nm ($60\,mW\,cm^{-2}$) for 3 min, U87MG cells ($2 \times 10^4$) were plated on a 35-mm confocal dish (SPL, Korea). When the cells were plated on the dish, they were treated with **KUP-1** and **O$_2$@ KUP-1** in SF media. After incubation for 12 h, the cells were rinsed twice with $1 \times$ PBS and irradiated with SF media at 530 nm. The cells on SF media were irradiated at 530 nm ($60\,mW\,cm^{-2}$) for 3 min, and the cells were incubated for 30 h at 37 °C with 5% $CO_2$. The frequency of live cells was determined according to the manufacturer's protocol (Cat #R37601, Thermo Fisher, USA).

## Lactate dehydrogenase assay in the presence of KUP-1 and O$_2$@ KUP-1
The U87MG cells were seeded into 96-well plates for 24 h at 37 °C (5% $CO_2$). To evaluate LDH release (Cat #C20300, Thermo Fisher, USA) in supernatants, the assay was conducted according to the manufacturer's protocols after incubation with **KUP-1** ($0.4\,mg\,mL^{-1}$) and **O$_2$@KUP-1** ($0.4\,mg\,mL^{-1}$) for 30 h. The release of LDH was measured by measuring the absorbance at 492 nm. Irradiation condition: 530 nm, $60\,mW\,cm^{-2}$, and 3 min.

## Hemolysis tests for KUP system
Blood was extracted from the hearts of mice anesthetized with isoflurane. At 4 °C, whole blood was centrifuged at 1.4 rcf to obtain RBCs. The purified RBCs were treated with **KUP-1**, **O$_2$@KUP-1**, **KUP-2**, and **O$_2$@KUP-2** (concentration: $0.4\,mg\,mL^{-1}$). The **KUP-1** and **O$_2$@KUP-1** samples were then exposed to photoirradiation at 530 nm ($60\,mW\,cm^{-2}$) for 3 min. The **KUP-2** and **O$_2$@KUP-2** samples were exposed to

photoirradiation at 660 nm (75 mW cm$^{-2}$) for 5 min. After incubation for 1 h at 37 °C, the prepared samples were centrifuged at 3000 rcf at 4 °C. The absorbance of the supernatant was measured at 492 nm.

## Animal

BALB/c mice (male, 5 weeks old) were obtained commercially from DBL (Gyeonggi-do, Republic of Korea). Five mice were randomly divided and housed per cage (20 × 26 × 13 cm) with free food and water intake in the room under a 12 h light/dark cycle (ambient temperature: 23 ± 1 °C, relative humidity: 60 ± 10%). All experiments performed with mice were carried out in accordance with the National Institute of Health Guide for the Care and Use of Laboratory Animals (NIH publications no. 80-23, revised 1996) and protocols approved by the Institutional Animal Care and Use Committee of Kyung Hee University for each experiment (KHUASP-(SE)-19-002 and KHUASP(SE)-19-003) and institutional guidelines (assigned no. 2015-020).

## Fabrication and blood test of U87MG-xenograft model

**Fabrication.** The implantation of U87MG cells was performed using Matrigel Membrane Matrix (Cat. No. 354234, Corning™, USA). Briefly, each mouse (BALB/c mice; 6–7 weeks) anesthetized with isoflurane has implanted a mixture with $5 \times 10^6$ U87MG cells (subcutaneous injection on back) and Matrigel Membrane Matrix as 1:1 ratio (injected volume: 50 µL). After GBM implantation, the mice were recovered in a constant temperature chamber (30 °C) and placed in individual cages.

**Photodynamic therapy.** The groups ($n = 5$ per group) were used to evaluate the effect of PDT. The in vivo photodynamic therapy was carried out using the PBS (control), **KUP-1**, **KUP-2**, **O$_2$@KUP-1**, and **O$_2$@KUP-2** in the GBM-xenograft model during the 16 days. The irradiation process (405 nm, 75 mW cm$^{-2}$, 3 min) was performed 1 day after administration (8 mpk) of **KUP-1**, **KUP-2**, **O$_2$@KUP-1**, and **O$_2$@KUP-2**. This experiment was repeatedly performed with independent conditions.

**Measurement of body weight and tumor size.** The GBM xenograft mice were weighed every 3–4 days until 16 days. The tumor size was measured by using a caliper. The size was calculated using a standard formula, such as $W$ (width) × $D$ (depth) × $H$ (height), to assume the volume of the tumor.

**Blood test.** At the point of 16 days, the blood of GBM xenograft mice was extracted from the hearts of mice anesthetized with isoflurane. At 4 °C, whole blood was centrifuged at 3000 rcf to obtain the serum protein. The prepared samples were kept at −21 °C and then melted at 4 °C for 1 h before being used. The protein mapping analysis (Cat. No. ARY006, Biotechne R&D Systems, USA) was performed at the manufacturer's recommendation. Briefly, the protein in blood was quantified using BCA (Cat. No. PI23227, Thermo Scientific™, USA) assay and then was loaded on a membrane coated by some antibodies for 1 h at 25 °C after washing and blocking the membrane. As a final step, the reaction with the antibody mixture was incubated overnight (about 12 h) at 25 °C. The antigen–antibody reaction on the membrane was measured by washing and reacting with enhanced chemiluminescence (ECL) solution in dark conditions.

## Fluorescence tissue imaging system (FTIS)

The U87MG-xenograft mice were measured by VISQUE InVivo Elite (Vieworks Co., Ltd., Republic of Korea) after administration with PBS, **KUP-1**, and **O$_2$@KUP-1** for 24 h. The concentration of administration was 8 mpk. The images were obtained 1 day after administration via subcutaneous injection under the tumor. [Setting information] zoom: 1×, focus: 109 steps, iris: F2.8, filter: excitation (540–569 nm), emission: (692–742 nm), light intensity: middle, exposure: 0.5 s, HDR: low gain mode.

## Statistics analysis

The data were expressed as the mean ± standard error of the mean (SEM) unless stated otherwise. Statistical comparisons of the toxicity analyses were performed using two-way ANOVA. For all test significance is defined as ****$P < 0.0001$; ***$P < 0.001$; **$P < 0.01$; *$P < 0.05$; $P$ values > 0.05 was considered not statistically significant (n.s.). Statistical tests were conducted using GraphPad InStat version 8.0.1 (GraphPad Software, La Jolla, CA, USA).

## Reporting summary

Further information on research design is available in the Nature Portfolio Reporting Summary linked to this article.

## Data availability

The authors declare that the data supporting the findings of this study are available within the Article and its Supplementary Information or from the corresponding author upon request. Source data are provided with this paper.

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

## Acknowledgements

This work was supported by the National Research Foundation of Korea funded by the Ministry of Science and ICT (NRF-2021R1A2B5B03086313 and 2019R1A6A1A11044070 for C.S.H.; CRI project no. 2018R1A3B1052702 for J.S.K.; NRF-2019R1A6A1A10073079 for J.Y.L.; NRF-2018R1A6A1A03025124 and 2022R1F1A1069954 for D.K.). This research was supported by a grant from the Korea Health Technology R&D Project through the Korea Health Industry Development Institute (KHIDI), funded by the Ministry of Health & Welfare, Republic of Korea (HI21C0239). We appreciate Prof. Kwangyeol Lee for the TEM measurements and Dr. Hyung Jong Kim and Prof. Dong Hoon Choi for the CV measurements. We appreciate the support from the KISTI

supercomputing center through the strategic support program for the supercomputing application research No. KSC-2021-CRE-0304 for J.Y.L. We appreciate BioRender.com. for image creation.

## Author contributions

D.W.K. synthesized and characterized the materials. J.S. implemented the solution tests using a laser. The project was designed by them. J.H.L. performed the computational calculations. J.M.A. performed the biological tests. The four authors mentioned above wrote the manuscript. Y.K. performed the TRPL experiments. J.H.K. performed the EPR measurements. M.S.J. contributed to the data discussion. S.P., D.K., J.Y.L., J.S.K., and C.S.H. edited and supervised the project.

## Competing interests

The authors declare no competing interests.
