## [Peer Review File · Nature Communications]

Wavelength engineerable porous organic polymer photosensitizers with protonation triggered ROS generationReviewers' Comments:

Reviewer #1:

Remarks to the Author:

This manuscript reports the preparation of the porous organic materials, and production of ROS. The ability of production of ROS is controlled by protonation/deprotonation. Although the results are interesting, I suggest a specialized journal is more suitable for this manuscript.

Some discussions in the manuscript need to be revised. For instance, 'electron relaxation..', this is not an appropriate photochemistry term, the excited state of an organic molecule is not only related to electron, much more other factors are involved, such as geometry, electron spin, etc.

'The ISC rate between S1 and T1 is derived by the reversed ISC', this is confusing, what not to calculate the forward ISC, i.e. from S1 to T1 state?

It is necessary to conform the formation of triplet state (i.e. ISC) of the materials, for instance, by nanosecond transient absorption spectra (laser flash photolysis).

Reviewer #2:

Remarks to the Author:

The authors synthesized two imidazoline-based porous organic polymers (POPs), namely KUP-1 and the extended version KUP-2 via a one-pot reaction. These POPs were well characterized with a range of spectroscopic and physical methods. The photophysical properties, ROS generation mechanism, and in vitro photodynamic activities of these POPs were also studied. Although this work reports the first imidazoline-based photosensitizers, there are quite a number examples of POP-based photosensitizers (e.g. *J. Mater. Chem. A* 2016, 4, 18677; *Angew. Chem. Int. Ed.* 2019, 58, 3062; *Chem. Commun.* 2021, 57, 6875). In addition, some of the studies and explanations are not clear. I think the work is not of sufficient novelty to meet the stringent requirement of this journal. Some specific comments are given below:

- (1) As shown in Fig. S15, KUP-2 showed strong absorption in 250-500 nm and very weak absorption in 600-800 nm. Could the authors explain why the singlet oxygen generation efficiency of KUP-2 is higher upon irradiation at a longer wavelength (660 or 808 nm) compared to excitation at 430 nm (Fig. S18)? Why is it different from that of KUP-1?
- (2) In p.8, the role of 2,2,6,6-tetramethylpiperidine (TEMP) and 5,5-dimethyl-1-pyrroline N-oxide (DMPO) in the EPR measurements should be mentioned.
- (3) The in vitro photocytotoxicity of KUP-2 and O2@KUP-2 should also be studied for comparison.
- (4) In p.13, why did the authors select the laser source at 530 nm with a power of 100 mW cm⁻² for the photocytotoxicity study? In the solution study of ABDA degradation (Fig. 2a and 2b), the solutions were excited at 430, 660, and 808 nm. In addition, the laser power used was only 1 mW cm⁻² and the authors claimed that their photosensitizers could generate ROS effectively even with a low-power laser irradiation compared to other conventional photosensitizers (p.7). It is suggested to study the in vitro photocytotoxicity with a lower laser power of 1 mW cm⁻², instead of 100 mW cm⁻², in order to support the claim.
- (5) The cells were incubated with KUP-1 and O2@KUP-1 for 12 h. Why was a long incubation time required? Would the oxygen content in O2@KUP-1 drop significantly during the long incubation time?
- (6) The authors claimed that the cell death underwent an 1O2-induced apoptosis pathway. More experiments are required to support this claim.
- (7) Besides the calculation, the intracellular generation of different ROS through type-I and type-II mechanisms should also be studied with specific ROS probes.
- (8) In vivo photodynamic therapy should also be performed to examine the therapeutic effect of the POPs.

Reviewer #3:

Remarks to the Author:

The authors report an imidazoline-based porous organic photosensitizer (KUP-1) and its extended version (KUP-2), to be used in ROS-utilizing applications. Both photosensitizers were prepared by a cost-effective and scalable one-pot reaction. Interestingly, these have good wettability and dispersibility in water because of the charged component of the structure, and they can load oxygen into their pores by virtue of their porosity and positive surface charge. Once they are irradiated with light, the polymers can effectively generate ROS containing 1O_2 and $O_2^{\bullet-}$, which follow both ROS generation mechanism types I and II, despite the absence of heavy atoms, being the wavelength for ROS generation adjustable in the visible range. These are the noteworthy results.

The authors applied computational methods to this polymeric system, along with experimental evidence, to understand the mechanism of ROS generation, which elucidates the intersystem crossing (ISC) dominance by polymerization. Furthermore, the biocompatibility of the photosensitizers was confirmed through biological experiments. I agree with the authors that these photosensitizing systems, with tunable excitation wavelengths, could provide a potential basis for designing nanoscale porous organic photosensitizers for a variety of ROS-enabled applications.

In my opinion, the manuscript is interesting, but not enough for recommending the acceptance of the paper in Nature Commun. The data provided support in part the conclusions and claims. There is enough detail provided in the methods for the work to be reproduced. The work is well performed, the writing is clear and concise, but the paper does not have great importance and outstanding novelty. The work significance to the field is average. This manuscript should be more appropriate for another more specialized journal.

[Reply to the referees]

Reply to the referee 1:

This manuscript reports the preparation of the porous organic materials, and production of ROS. The ability of production of ROS is controlled by protonation/deprotonation. Although the results are interesting, I suggest a specialized journal is more suitable for this manuscript.

Some discussions in the manuscript need to be revised. For instance, 'electron relaxation.', this is not an appropriate photochemistry term, the excited state of an organic molecule is not only related to electron, much more other factors are involved, such as geometry, electron spin, etc.

<Previous version of the manuscript>

After the electrons are excited by the irradiated light, electron relaxation is performed through non-radiative decay, including internal conversion (IC) and ISC processes.

<New version of the manuscript>

After the photosensitizer is excited to the S₁ state by the irradiated light, the excited system can be relaxed through a non-radiative process such as internal conversion (IC) and ISC processes.

Response: We thank the referee for pointing out this inconsistency. Considering other factors related to the excited state, we changed the expression; “electron relaxation” into “excited system can be relaxed”.

'The ISC rate between S₁ and T₁ is derived by the reversed ISC', this is confusing, what not to calculate the forward ISC, i.e. from S₁ to T₁ state?

It is necessary to conform the formation of triplet state (i.e. ISC) of the materials, for instance, by nanosecond transient absorption spectra (laser flash photolysis).

<Previous version of the manuscript>

The ISC rate between S₁ and T₁ is derived by the reverse ISC (RISC) rate, which is computed from the semiempirical Marcus theory (Eq. 1).⁴⁰

$$k_{\text{ISC}} = \frac{4\pi^2}{h} \frac{\langle S_1 | H_{\text{SOC}} | T_1 \rangle^2}{\sqrt{4\pi\lambda k_B T}} \exp\left(-\frac{(\text{MECP})^2}{k_B T}\right) \quad (1a)$$

$$\text{MECP} = \frac{(\Delta E_{\text{ST}} - \lambda)^2}{4\lambda} \quad (1b)$$

<New version of the manuscript>

The ISC rate between S₁ and T₁ was computed from the semiempirical Marcus theory (Eq. 1).⁴⁰

$$k_{\text{ISC}} = \frac{4\pi^2 \langle S_1 | H_{\text{SOC}} | T_1 \rangle^2}{h \sqrt{4\pi\lambda k_B T}} \exp\left(-\frac{(\text{MECP})^2}{k_B T}\right) \quad (1a)$$

$$\text{MECP} = \frac{(\Delta E_{\text{ST}} - \lambda)^2}{4\lambda} \quad (1b)$$

Response: Marcus theory was developed to explain the rate of electron transfer and gives a pre-exponential term affected by electronic coupling, reorganization energy, and temperature. In general, Marcus theory is applied to calculate the up-conversion rate between S₁ and T₁ states, which is why we used the expression “The ISC rate between S₁ and T₁ is derived by the reverse ISC (RISC) rate”. However, these factors are the same in the processes of ISC and RISC. Thus, it can be changed to the simple new version.

Reply to the referee 2:

The authors synthesized two imidazoline-based porous organic polymers (POPs), namely KUP-1 and the extended version KUP-2 via a one-pot reaction. These POPs were well characterized with a range of spectroscopic and physical methods. The photophysical properties, ROS generation mechanism, and in vitro photodynamic activities of these POPs were also studied. Although this work reports the first imidazoline-based photosensitizers, there are quite a number examples of POP-based photosensitizers (e.g. J. Mater. Chem. A 2016, 4, 18677; Angew. Chem. Int. Ed. 2019, 58, 3062; Chem. Commun. 2021, 57, 6875). In addition, some of the studies and explanations are not clear. I think the work is not of sufficient novelty to meet the stringent requirement of this journal. Some specific comments are given below:

(1) As shown in Fig. S15, KUP-2 showed strong absorption in 250-500 nm and very weak absorption in 600-800 nm. Could the authors explain why the singlet oxygen generation efficiency of KUP-2 is higher upon irradiation at a longer wavelength (660 or 808 nm) compared to excitation at 430 nm (Fig. S18)? Why is it different from that of KUP-1?

Response: We thank the reviewer for pointing this out. Unlike solvent-soluble small molecular photosensitizers, the KUP system is a polymeric photosensitizer that is barely soluble in various solvents. Therefore, we measured the solid-state UV (Fig.S15) to get

a hint about the absorbance area of KUP-1 and KUP-2. However, the absorption spectrum obtained from solid-state and solution-state UV sometimes can have different behaviors in absorption spectra due to particle aggregation and dispersity by solvent-affinity (*J. Mater. Chem. C*, **5**, 12349-12353, (2017). DOI: 10.1039/C7TC04621A; *Chem. Sci.*, **9**, 1596-1603, (2018). DOI: 10.1039/C7SC03739B; *Photochem. Photobiol. B, Biol.*, **158**, 212–218, (2016). DOI: 10.1016/j.jphotobiol.2016.03.004), A wavelength where the maximum efficiency of ROS generation is generated was observed by changing the actual laser wavelength range, and KUP-2 produced ROS at a longer wavelength than KUP-1 (Fig. S16 and S18).

(2) In p.8, the role of 2,2,6,6-tetramethylpiperidine (TEMP) and 5,5-dimethyl-1-pyrroline N-oxide (DMPO) in the EPR measurements should be mentioned.

<Previous version of the manuscript>

To determine the different ROS types generated by the KUP system under light irradiation, electron paramagnetic resonance (EPR) measurements were performed with 2,2,6,6-tetramethylpiperidine (TEMP) and 5,5-dimethyl-1-pyrroline N-oxide (DMPO) as a ROS generation indicator (Figs. 3e and 3f).

<New version of the manuscript>

To determine the different ROS types generated by the KUP system under light irradiation, electron paramagnetic resonance (EPR) measurements were performed with 2,2,6,6-tetramethylpiperidine (TEMP) and 5,5-dimethyl-1-pyrroline N-oxide (DMPO) as a $^1\text{O}_2$ and $\text{O}_2^{\cdot-}$ generation indicator, respectively (Figs. 3e and 3f).

Response: We thank the reviewer for this suggestion. We changed the expression; “as a ROS generation indicator” into “as a $^1\text{O}_2$ and $\text{O}_2^{\cdot-}$ generation indicator, respectively”.

(3) The in vitro photocytotoxicity of KUP-2 and O2@KUP-2 should also be studied for comparison.

<New version of the manuscript>

Fig. S29 The cytotoxicity ability assay of **KUP-2** and **O₂@KUP-2** after irradiation at 660 nm. The U87MG (glioblastoma), C6 (glioblastoma), and HeLa (cervical cancer) cell lines were used. The incubation of **KUP-1** and **O₂@KUP-1** was fixed at 12 h for being uptaken into the cytosol. The measurement of toxicity was performed after incubation for 24 h.

Fig. S32 Tumor sizes and body weights of the mice in each tested group recorded during treatment with/without 660 nm irradiation at the end-point ($n = 5$, number of mice). (L): laser.

“Unlike **KUP-1** and **O₂@KUP-1**, we could not observe the positive results using **KUP-2** and **O₂@KUP-2** in diverse cell lines (C6, HeLa, and U87MG) as shown in Fig. S29.”

“The difference in our results between in vitro and in vivo might be caused by the limitations of 2D cultures, such as cell shape with forced polarity and lack of immune cells. The plate of 2D culture was modified to force polarity to make the cell easier to attach to the plate. The modified plate drives to change the cellular morphology, unlike in vivo, although it helps the cell to attach to the plate. Also, the tumor microenvironment consists of diverse cell types, even containing the normal cell and immune cells, which all have a role in the function of the tumor.[ref: DOI:10.3390/ijms19010181] Thus, many drugs have worked differently in vitro and in vivo. Based on these things, **KUP-2** and **O₂@KUP-2** might work near the tumor site

connecting with diverse cells in 3D structure.”

Response: We thank the reviewer for pointing this out. Following the reviewer’s concern, we evaluated the in vitro test and additional in vivo analysis using KUP-2 and O₂@KUP-2. Unfortunately, we were not able to obtain the positive results of KUP-2 and O₂@KUP-2 using some cell lines (C6, HeLa, and U87MG). However, it was possible to observe the positive ability for photo-cytotoxicity of KUP-2 and O₂@KUP-2 in vivo tumor model without any big issues. As far as we are concerned, the difference in our results between in vitro and in vivo might be caused by the limitations of 2D cultures, such as cell shape with forced polarity and lack of immune cells. The plate of 2D culture was modified to force polarity to make the cell easier to attach to the plate. The modified plate drives to change the cellular morphology, unlike in vivo, although it helps the cell to attach to the plate. Also, the tumor microenvironment consists of diverse cell types, even containing the normal cell and immune cells, which all have a role in the function of the tumor (*Int. J. Mol. Sci.* **19**, 181, (2018). DOI:10.3390/ijms19010181). Thus, many drugs have worked differently in vitro and in vivo. Based on these observations, KUP-2 and O₂@KUP-2 might work near the tumor site connecting with diverse cells in a 3D structure. Additionally, the results were added in Fig. S29 and Fig. S32, and related sentences were described in the part of the result.

(4) In p.13, why did the authors select the laser source at 530 nm with a power of 100 mW cm⁻² for the photocytotoxicity study? In the solution study of ABDA degradation (Fig. 2a and 2b), the solutions were excited at 430, 660, and 808 nm. In addition, the laser power used was only 1 mW cm⁻² and the authors claimed that their photosensitizers could generate ROS effectively even with a low-power laser irradiation compared to other conventional photosensitizers (p.7). It is suggested to study the in vitro photocytotoxicity with a lower laser power of 1 mW cm⁻², instead of 100 mW cm⁻², in order to support the claim.

Response: We thank the reviewer for pointing this out. After receiving the comment, we measured the power intensity of all lasers and lamp sources again using a power meter device, and confirmed that the value of the LED lamp used to measure in vitro was incorrect (Xenon lamp for solution-based analysis: 1 mW cm⁻², Lamp for in vitro analysis: 60 mW cm⁻², and laser for in vivo analysis: 75 mW cm⁻²). Although the in vitro photo-cytotoxicity was not conducted at a lower laser power of 1 mW cm⁻² due to the instrumental absence of a light source with the lower power, the power intensities of all light sources used in the experiments were lower than that of used by conventional

photosensitizers in therapeutic and photocatalytic applications (*Chem. Eng. J.* **413**, 127412, (2021). DOI: 10.1016/j.ccej.2020.127412; *Chem. Soc. Rev.* **48**, 2053-2108, (2019). DOI: 10.1039/C8CS00618K; *J. Am. Chem. Soc.* **139**, 8705-8709, (2017). DOI: 10.1021/jacs.7b04141).

(5) The cells were incubated with KUP-1 and O₂@KUP-1 for 12 h. Why was a long incubation time required? Would the oxygen content in O₂@KUP-1 drop significantly during the long incubation time?

<New version of the manuscript>

Fig. 2 Synthesis and structural characterization of porous organic photosensitizers. **a**, Schematic illustration of the porous organic polymeric photocatalytic ROS generation

system, KUP system, and underlying imidazolium-based porous organic photosensitizers to engineering operating wavelength for maximum efficiency. **b**, XPS survey scan and **c**, narrow scan data of N1s peak of **KUP-1**. **d**, Solid-state ^{15}N NMR data of **KUP-1**. **e**, CO_2 isotherms of porous organic polymers at 195 K. **f**, Surface charge distributions of **KUP-1** and **O₂@KUP-1**. **g**, TEM images ($\times 63,000$ and $\times 250,000$; inset) of the prepared **KUP-1**. The scale bars are 200 nm and 50 nm, respectively.

“In addition, we measured the zeta potential after incubation of **O₂@KUP-1** in an aqueous solution to check the O_2 transport performance of the KUP system. As a result, the surface charge of **O₂@KUP-1** is still kept even after 24 h, indicating that oxygen content in **O₂@KUP-1** will not drop significantly during O_2 delivery (Fig. 2f).”

Response: We thank the reviewer for pointing this out. We assumed that KUP-1 and **O₂@KUP-1** might quite stay near the tumor before being uptaken by tumor cells due to the complex blood vessels in the tumor microenvironment. To confirm if the oxygen was laid off, we measured the zeta potential after incubation of **O₂@KUP-1** in an aqueous solution. As a result, the surface charge of KUP-1 is still kept even after 24 h, indicating that oxygen content in **O₂@KUP-1** will not drop significantly during our incubation time (12 h). The results were added in Fig. 2f and Fig. S2, and related sentences were described in the part of the result.

(6) The authors claimed that the cell death underwent an $^1\text{O}_2$ -induced apoptosis pathway. More experiments are required to support this claim.

<New version of the manuscript>

Fig. 5 *In vitro* and *in vivo* bioactivity of **KUP-1** and **O₂@KUP-1**. **a**, Cytotoxicity assays of **KUP-1** and **O₂@KUP-1** with/without the presence of 530 nm laser irradiation. **b**, Live-cell images after treatment with **KUP-1** and **O₂@KUP-1** under the photoirradiation at 530 nm. Scale bars: 200 μ m. **c**, Lactate dehydrogenase (LDH) assays after treatment of **KUP-1** and **O₂@KUP-1**. Significance represents two-way ANOVA result. **d**, Hemolysis tests for **KUP-1** and **O₂@KUP-1**. Neg: negative control (PBS), Pos: positive control; 0.1% (v/v) Triton X-100. 1: the group which **KUP-1** treats without irradiation, 2: the group which is treated by PBS with irradiation at 530 nm, 3: the group which is treated by **KUP-1** with irradiation at 530 nm, 4: the group which is treated by **O₂@KUP-1** with irradiation at 530 nm. An inset photograph is a supernatant of the damaged red blood cells. **e**, Schematic illustration for *in vivo* evaluation of **KUP-1** with PDT irradiation in GBM-xenograft model. **f**, Images of mouse condition after treatment of 1 \times PBS, **KUP-1** (8 mpk; mg kg⁻¹), and **O₂@KUP-1** (8 mpk) with photoirradiation for 13 days. **g**, FTIS images of the mouse with treatment of 1 \times PBS, **KUP-1** (8 mpk), and **O₂@KUP-1** (8 mpk). **h**, Tumor sizes and body weights of the mice in each tested group

recorded during treatment with/without 405 nm irradiation (75 mW cm⁻², 3 min) at the end-point. Red arrows indicate the PS treatment date. **i**, Profiler mouse cytokine assay using extracted blood from the GBM-xenograft model at the end-point. Blue box means cytokine proteins that are increased after irradiation at 405 nm. **j**, Protein expression mapping of **KUP-1** and **O₂@KUP-1**. The data was analyzed using **i**. The number of mice per group is 6; independent experiments with duplicates repeatedly obtained all data. All data were presented as mean ± S.E.M., **p<0.01, ***p<0.001, ns= non-significant. (L): laser irradiation.

Fig. S31 Images of mouse condition after treatment of 1×PBS, **KUP-1** (8 mpk; mg/kg), and **O₂@KUP-1** (8 mpk) with/without photo-induced treatment for 13 days.

Response: We thank the reviewer for pointing this out. To address the reviewer’s concern, we performed the protein profiling assay using the in vivo model. As shown in Fig. 5i-j, the photo-toxicity of O₂@KUP-1 was induced mainly by I-TAC, C5/C5a, and CD54, which are released from the immune cells near the tumor microenvironment. Thus, we think the KUP-1 and O₂@KUP-1 could release the ROS and activate the immune cell line near the tumor microenvironment to reduce the tumor region. The results were added in Fig. 5 and Fig. S31, and related sentences were described in the part of the result.

(7) Besides the calculation, the intracellular generation of different ROS through type-I and type-II mechanisms should also be studied with specific ROS probes.

Response: We thank the reviewer for this suggestion. In addition to question (2), EPR experiments were measured to determine the type of ROS that occurs after photoirradiation. ABDA and DMPO used in the experiment were ¹O₂ (Type-II ROS) and O₂^{•-} (Type-II ROS) indicators, respectively. Therefore, we changed the expression; “as a ROS generation indicator” into “as a ¹O₂ and O₂^{•-} generation indicator, respectively”.

(8) In vivo photodynamic therapy should also be performed to examine the therapeutic effect of the POPs.

Response: We thank the reviewer for pointing this out. As the reviewer commented, we additionally performed the in vivo photodynamic therapy test in the mouse and confirmed the therapy effect. The data was updated in Fig. 5, Fig. S31, and Fig. S32, and sentences related to the data were described in the part of the result.

Reply to the referee 3:

The authors report an imidazoline-based porous organic photosensitizer (KUP-1) and its extended version (KUP-2), to be used in ROS-utilizing applications. Both photosensitizers were prepared by a cost-effective and scalable one-pot reaction. Interestingly, these have good wettability and dispersibility in water because of the charged component of the structure, and they can load oxygen into their pores by virtue of their porosity and positive surface charge. Once they are irradiated with light, the polymers can effectively generate ROS containing $^1\text{O}_2$ and $\text{O}_2^{\cdot-}$, which follow both ROS generation mechanism types I and II, despite the absence of heavy atoms, being the wavelength for ROS generation adjustable in the visible range. These are the noteworthy results.

The authors applied computational methods to this polymeric system, along with experimental evidence, to understand the mechanism of ROS generation, which elucidates the intersystem crossing (ISC) dominance by polymerization. Furthermore, the biocompatibility of the photosensitizers was confirmed through biological experiments. I agree with the authors that these photosensitizing systems, with tunable excitation wavelengths, could provide a potential basis for designing nanoscale porous organic photosensitizers for a variety of ROS-enabled applications.

In my opinion, the manuscript is interesting, but not enough for recommending the acceptance of the paper in Nature Commun. The data provided support in part the conclusions and claims. There is enough detail provided in the methods for the work to be reproduced. The work is well performed, the writing is clear and concise, but the paper does not have great importance and outstanding novelty. The work significance to the field is average. This manuscript should be more appropriate for another more specialized journal

Response: We thank the referee for the valuable comments. After given that turn of comments, we decided to basically start over, both to strengthen the evidence from top to bottom and to make clear why the work was important. In our view, the weakness of the original submission was that it was limited to cell-based studies and only speculated

the photosensitizing mechanism of the KUP system by the calculation-related utilities, not as experimental evidence. As a result, we believe the current work does represent a major step forward. In brief, after a year of dedicated effort across multiple laboratories, we have been able to show for the first time that an appropriately designed protonated-exclusive porous organic polymeric system engineered its active excitation wavelength for optimal photocatalytic ROS generation in both in vitro and in vivo assays. In addition, we made up for the photophysical evidence to elucidate the ROS generation mechanism from time-resolved photoluminescence experiments. This is noteworthy that controlling the wavelength of photosensitizer corresponding to operating conditions is critical if we are to achieve systemic control over what yields an absolute effect under various conditions and environments. The approach we detail here also offers the possibility of expanding the utility of the photosensitizers. We are of the opinion that demonstrating how a relatively small rationalized change in monomer design has major implications to its properties is of interest to the readership of a high-profile general chemistry journal such as *Nature Communications*.

Reviewers' Comments:

Reviewer #1:

Remarks to the Author:

The revision is satisfactory, I recommend acceptance of the manuscript in its current form.

Reviewer #2:

Remarks to the Author:

The authors have satisfactorily addressed my previous comments. However, there are some additional problems as listed below:

- (1) Figs. S21 (Tauc plot) and S22 (CV data) should be interchanged in the SI file.
- (2) In p.15, line 401, there is a typo for O2@KUP-1.
- (3) In Fig. 5g, the whole-body fluorescence of the mouse being injected with PBS was very bright and comparable with that of the mice being injected with KUP-1 or O2@KUP-1. Can this be explained?
- (4) In Fig. 5h, there are two arrows indicating the PS treatment date (as stated in the caption). However, a single dose of PS is described in the text (p.17-18, line 433-436).
- (5) The procedure for the in vivo photodynamic therapy is not mentioned in the Methods section.

Reviewer #3:

Remarks to the Author:

After an initial decision, the authors have submitted a revised version. I have carefully read the manuscript and the response to the referees, and it is my opinion that the authors have made a big effort to improve the first version of their manuscript. In general, adequate answers have been given to their questions. The work is well done, and the literature is well up to date. However, my opinion is the same as before. The paper does not have outstanding novelty. The work's significance to the field is average. This manuscript should be more appropriate for another more specialized journal.

[Reply to the referees]

Reviewer #1 (Remarks to the Author):

The revision is satisfactory, I recommend acceptance of the manuscript in its current form.

Response: We thank the referee for the valuable comment.

Reviewer #2 (Remarks to the Author):

The authors have satisfactorily addressed my previous comments. However, there are some additional problems as listed below:

Response: We thank the referee for the encouraging remarks.

(1) Figs. S21 (Tauc plot) and S22 (CV data) should be interchanged in the SI file.

Response: We thank the reviewer for pointing this out. As the reviewer commented, we interchanged the SI figures.

(2) In p.15, line 401, there is a typo for O₂@KUP-1.

Response: We thank the reviewer for pointing this out. We corrected the typo error.

(3) In Fig. 5g, the whole-body fluorescence of the mouse being injected with PBS was very bright and comparable with that of the mice being injected with KUP-1 or O₂@KUP-1. Can this be explained?

Response: We thank the reviewer for this suggestion. The mouse model (balb/c) was fabricated by subcutaneous inoculation of U87MG on dorsal. The growth of inoculated U87MG is generated on the skin that has thin hair with autofluorescence under the excitation (540–569 nm). Thus, the result of mouse being injected with PBS might be caused by autofluorescence.

(4) In Fig. 5h, there are two arrows indicating the PS treatment date (as stated in the caption). However, a single dose of PS is described in the text (p.17-18, line 433-436).

Response: We thank the reviewer for pointing this out. To address the reviewer's concern, we revised the text to reduce the misunderstanding from "Based on this result, we performed the irradiation process 1 day after treating KUP-1 or O₂@KUP-1" to "Based on this result, we performed the irradiation process in 1 day after treating KUP-1 or O₂@KUP-1 (2 cycles)".

(5) The procedure for the in vivo photodynamic therapy is not mentioned in the Methods section.

Response: We thank the reviewer for pointing this out. As the reviewer commented, we additionally updated the methods section for in vivo PDT as a below.

Fabrication and blood test of U87MG-xenograft model.

(ii) Photodynamic therapy: The groups (n=5 per group) were used to evaluate the effect of PDT. The in vivo photodynamic therapy were carried out using the PBS (control), **KUP-1**, **KUP-2**, **O₂@KUP-1**, and **O₂@KUP-2** in GBM-xenograft model during the 16 days. The irradiation process (405 nm, 75 mW cm⁻², 3 min) was performed in 1 day after administration (8 mpk) of **KUP-1**, **KUP-2**, **O₂@KUP-1**, and **O₂@KUP-2**. This experiment was repeatedly performed with independent conditions.

Reviewer #3 (Remarks to the Author):

After an initial decision, the authors have submitted a revised version. I have carefully read the manuscript and the response to the referees, and it is my opinion that the authors have made a big effort to improve the first version of their manuscript. In general, adequate answers have been given to their questions. The work is well done, and the literature is well up to date. However, my opinion is the same as before. The paper does not have outstanding novelty. The work significance to the field is average. This manuscript should be more appropriate for another more specialized journal.

Response: We thank the referee for the valuable comments. We believe the current work does represent a major step forward. Indeed, some current porous material-based photosensitizers have been reported with a good photosensitizing. But the key finding of this work lies in rationalizing and validating the ROS generation system and antitumor efficiency from a mechanistic point of view. Furthermore, these mechanistic insights are likely to be translated and applied to the general advancement of wavelength engineerable porous organic polymer photosensitizer design. We are of the opinion that demonstrating how a relatively rationalized small change in monomer has major implications to its properties is of interest to the readership of a high-profile journal such as *Nature Communications*.